# Non-canonical NF-κB signaling limits the tolerogenic β-catenin-Raldh2 axis in gut dendritic cells to exacerbate intestinal pathologies

Alvina Deka [1,5], Naveen Kumar[1,5], Swapnava Basu[1], Meenakshi Chawla[1], Namrata Bhattacharya[2,3], Sk Asif Ali [1], Bhawna[1], Upasna Madan[4], Shakti Kumar[4], Bhabatosh Das[4], Debarka Sengupta [2], Amit Awasthi[4] & Soumen Basak [1]✉

## Abstract

**Dendritic cell (DC) dysfunction is known to exacerbate intestinal pathologies, but the mechanisms compromising DC-mediated immune regulation in this context remain unclear. Here, we show that intestinal dendritic cells from a mouse model of experimental colitis exhibit significant levels of noncanonical NF-κB signaling, which activates the RelB:p52 heterodimer. Genetic inactivation of this pathway in DCs alleviates intestinal pathologies in mice suffering from colitis. Deficiency of RelB:p52 diminishes transcription of Axin1, a critical component of the β-catenin destruction complex, reinforcing β-catenin-dependent expression of Raldh2, which imparts tolerogenic DC attributes by promoting retinoic acid synthesis. DC-specific impairment of noncanonical NF-κB signaling leads to increased colonic numbers of Tregs and IgA+ B cells, which promote luminal IgA production and foster eubiosis. Experimentally introduced β-catenin haploinsufficiency in DCs with deficient noncanonical NF-κB signaling moderates Raldh2 activity, reinstating colitogenic sensitivity in mice. Finally, inflammatory bowel-disease patients also display a deleterious noncanonical NF-κB signaling signature in intestinal DCs. In sum, we establish how noncanonical NF-κB signaling in dendritic cells can subvert retinoic acid synthesis to fuel intestinal inflammation.**

**Keywords** IgA; Inflammation; Inflammatory Bowel Disease; RelB; Retinoic Acid
**Subject Categories** Digestive System; Immunology; Signal Transduction

## Introduction

DCs orchestrate the balance between protective immune responses against pathogens and tolerance toward commensal microbes in the intestine. Upon microbial sensing, activated DCs produce a myriad of immunogenic cytokines, including IL-6, IL-12, and IL-23, which promote IFNγ-secreting Th1 and IL-17-secreting Th17 cells. These effector T cells direct inflammatory responses, containing gut infections. DCs also produce tolerogenic cytokines, such as IL-10 and TGFβ. In addition, mucosal DCs synthesize retinoic acid (RA) from dietary vitamin A or retinol (Stagg, 2018). These immunomodulatory factors support the generation and maintenance of inflammation-suppressive FoxP3+ Tregs and promote their gut homing. Furthermore, DC-derived RA and IL-10 enhance immunoglobulin class switching, accumulating IgA-secreting plasma B cells in the intestine (Xu et al, 2007; Seo et al, 2013). While secretory IgA (sIgA) in the gut lumen protects from opportunistic gut pathogens, sIgA coating also shapes the gut microbiome in symbiosis with the host (Bunker et al, 2017; Nakajima et al, 2018). Thus, by calibrating mucosal responses, DCs ensure gut homeostasis.

Not surprisingly, distorted DC functions have been implicated in aberrant intestinal inflammation. Ulcerative colitis (UC), a form of IBD, was associated with a marked reduction in the CD103+ DC subset in the intestine (Magnusson et al, 2016). Colonic DCs from IBD patients or mice subjected to experimental colitis were inept at synthesizing TGFβ and RA (Collins et al, 2011; Magnusson et al, 2016). These DCs were also less proficient in generating Tregs and instead promoted Th1 and Th17 cells. Accordingly, DC-specific ablation of RXRα, an RA-activated transcription factor, aggravated colitis in mice by depleting intestinal Tregs (Manoharan et al, 2023). Similarly, CD137 signaling restricted colitis in mice by upregulating RA synthesis in DCs that altered T-cell homeostasis in the intestine (Jin et al, 2020). Furthermore, vitamin A deficiency exacerbated colitis in mice by decreasing the colonic abundance of IgA-secreting cells (Okayasu et al, 2016). Notably, IgA-coated gut microbes from healthy individuals provided protection, while those from IBD patients sensitized mice to experimental colitis (Kau et al,

[1]Systems Immunology Laboratory, National Institute of Immunology, Aruna Asaf Ali Marg, New Delhi 110067, India. [2]Indraprastha Institute of Information Technology Delhi, New Delhi, India. [3]Australian Prostate Cancer Research Centre—Queensland, Institute of Health and Biomedical Innovation, School of Biomedical Sciences, Queensland University of Technology (QUT), Brisbane, QLD, Australia. [4]Translational Health Science and Technology Institute, Faridabad, Haryana, India. [5]These authors contributed equally: Alvina Deka, Naveen Kumar. ✉E-mail: sobasak@nii.ac.in

2015; Palm et al, 2014). What triggers such DC dysfunction in intestinal pathologies remains unclear.

The NF-κB system comprises two interlinked canonical and noncanonical signaling arms, which play important roles in the differentiation, survival, and functioning of DCs. In particular, the canonical NF-κB pathway mediates microbial signal-responsive nuclear translocation of RelA- and cRel-containing NF-κB transcription factors, which induce the expression of immunogenic cytokines, including IL-1β, IL-12, and IL-23, in DCs (Mukherjee et al, 2024). Previous studies suggested that the activation of canonical NF-κB signaling in DCs worsens experimental colitis in mice (Visekruna et al, 2015). On the other hand, the noncanonical NF-κB pathway is activated by a subset of TNFR superfamily members, including lymphotoxin-β receptor (LTβR), GM-CSFR, and CD40 (Sun, 2017). In this pathway, NF-κB inducing kinase (NIK)-dependent phosphorylation promotes processing of the NF-κB precursor protein p100 into the mature p52 subunit, leading to nuclear accumulation of RelB:p52. In a non-redundant manner to the canonical pathway, noncanonical NF-κB signaling in DCs induces the expression of IL-23, which drives Th17 responses (Shih et al, 2012; Hofmann et al, 2011). NIK function in DCs was implicated in the pathogenic Th17 response underlying epidermal damage in the mouse model of psoriasis (Huang et al, 2018). DC-intrinsic role of NIK and RelB also restricted Treg accumulation in the central nervous system in the mouse model of experimental autoimmune encephalomyelitis (EAE) (Andreas et al, 2019; Hofmann et al, 2011). In the intestine, NIK-dependent IL-23 expression by DCs was important for Th17-mediated immune controls of *Citrobacter rodentium* (Jie et al, 2018). As such, NIK in DCs promoted pIgR expression in IECs involving Th17-secreted IL-17 that maintained luminal IgA levels and a gut-protective microbiome. Interestingly, *LTBR* and *NFKB2* were linked in a genome-wide association to the susceptibility loci for IBD (Liu et al, 2015). We asked if noncanonical RelB:p52 NF-κB signaling perturbed DC functions in the inflamed gut.

Here, we report that human IBD and experimental colitis in mice are associated with intensified noncanonical NF-κB signaling in intestinal DCs. Genetic disruption of this pathway in DCs alleviated chemical-induced colitis in mice. Our mechanistic studies suggested that RelB:p52 transcriptionally upregulated the expression of Axin1, which tethers β-catenin to its destruction complex. Inactivation of noncanonical NF-κB signaling reinforced the β-catenin-mediated expression of Raldh2, which promotes tolerogenic RA synthesis by DCs (Manicassamy et al, 2010). Ablating *Relb* or *Nfkb2* in DCs not only increased the frequency of intestinal Tregs but also improved the abundance of colonic IgA⁺ B cells, which fostered luminal IgA and the gut microbiome. Finally, haploinsufficiency of β-catenin in DCs lacking noncanonical NF-κB signaling reinstated colitogenic sensitivity in the composite knockout mice. Taken together, we chart a novel crosstalk between the noncanonical NF-κB pathway and the tolerogenic β-catenin–Raldh2 axis in DCs in tuning intestinal inflammation.

## Results

### Experimental colitis in mice is associated with and exacerbated by noncanonical *Relb-Nfkb2* signaling in DCs

To decipher if intestinal pathologies are linked to noncanonical NF-κB signaling in DCs, we interrogated single-cell RNA-seq data generated using colon tissues from mice subjected to dextran sodium sulfate

(DSS)-induced acute colitis (Ho et al, 2021) (Appendix Fig. S1A). Based on the expression of prototypic macrophage- and DC-specific genes (Appendix Table S1), we segregated the intestinal mononuclear phagocyte (MNPs) population described in this dataset into macrophage and DC subsets (Fig. 1A). Our subsequent analyses revealed a relatively insignificant level of *Relb* and *Nfkb2* mRNAs in macrophages in comparison to DCs, suggesting that noncanonical NF-κB functions could be more relevant for DCs among intestinal MNPs (Fig. 1B). Accordingly, we focused on *Relb* or *Nfkb2* mRNA-expressing DCs in the gut (Fig. 1C; Appendix Fig. S1B). DSS treatment led to a rise in the abundance of mRNAs encoding these noncanonical NF-κB factors in intestinal DCs at day 6. We also observed an increase in the frequency of DCs expressing these mRNAs in the colitogenic gut (Fig. 1C; Appendix Fig. S1B). Based on a previously described bulk transcriptomics study involving RelB-deficient mouse DCs (Shih et al, 2012), we then deduced a panel of five top RelB-important genes—*Fgr*, *Top1*, *Crk*, *Gpd2*, and *Cript* —that were also less reliant on other NF-κB factors for their expressions. We determined the abundance of corresponding mRNAs as a surrogate of noncanonical NF-κB activity. We found DCs from mouse colon possessed detectable levels of *Fgr*, *Top1*, and *Cript* transcripts even prior to DSS treatment (Fig. 1D). Except for *Top1*, DSS treatment led to elevated expressions of all other genes in intestinal DCs that were accompanied by an increased frequency of DCs expressing these mRNAs (Fig. 1D). Our UMAP also captured the appearance of a distinct DC-population in the intestine of DSS-treated mice. Cataloguing DCs into classical DC1(cDC1) and cDC2 clarified that this population consisted of cDC2 (Appendix Fig. S1C and Appendix Table S1). However, we did not see discernable differences between these DC subsets with respect to the noncanonical NF-κB pathway engagement (Appendix Fig. S1D). These studies implied DC-intrinsic noncanonical NF-κB signaling in experimental colitis.

To demonstrate noncanonical NF-κB activation in intestinal DCs directly, we harvested DCs from gut-draining mesenteric lymph nodes (MLNs) as CD11c⁺CD64⁻MHCII^high cells and performed immunoblot analyses. Intestinal DCs from untreated mice displayed basal processing of p100 into p52, a hallmark of noncanonical NF-κB signaling (Fig. 1E; Appendix Fig. S1E). DSS treatment further augmented p100 processing in these DCs in mice. We then asked what could be stimulating the noncanonical NF-κB pathway in intestinal DCs. As such, LTβR and GM-CSFR both have been implicated in the noncanonical NF-κB activation in DCs (Jin et al, 2014; Pian et al, 2020). Our single-cell RNA-seq data analyses revealed a substantial expression of mRNAs encoding LTβR and GM-CSFR, alternately termed Csf2R, in intestinal DCs even from mice that were not challenged (Fig. 1C). As described earlier (Upadhyay and Fu, 2013), a significant fraction of B and T lymphocytes present in the gut expressed the lymphotoxin ligand Ltb, whose abundance was elevated upon DSS treatment (Fig. 1F). However, we did not find any representation of innate lymphoid cells in this dataset. Also, immune cells expressing Csf2, the cognate ligand for GM-CSFR, were less prevalent. Accordingly, we reasoned that LTβR stimulated noncanonical NF-κB signaling in intestinal DCs. Indeed, disruption of lymphotoxin-mediated LTβR engagement in mice using LTβR-IgG fusion protein downmodulated p100 processing to p52 in intestinal DCs (Fig. 1E; Appendix Fig. S1E). We concluded that intestinal DCs possessed a basally active noncanonical *Relb-Nfkb2* pathway, and that aberrant intestinal inflammation was associated with intensified noncanonical NF-κB signaling in these cells, presumably involving LTβR engagement.

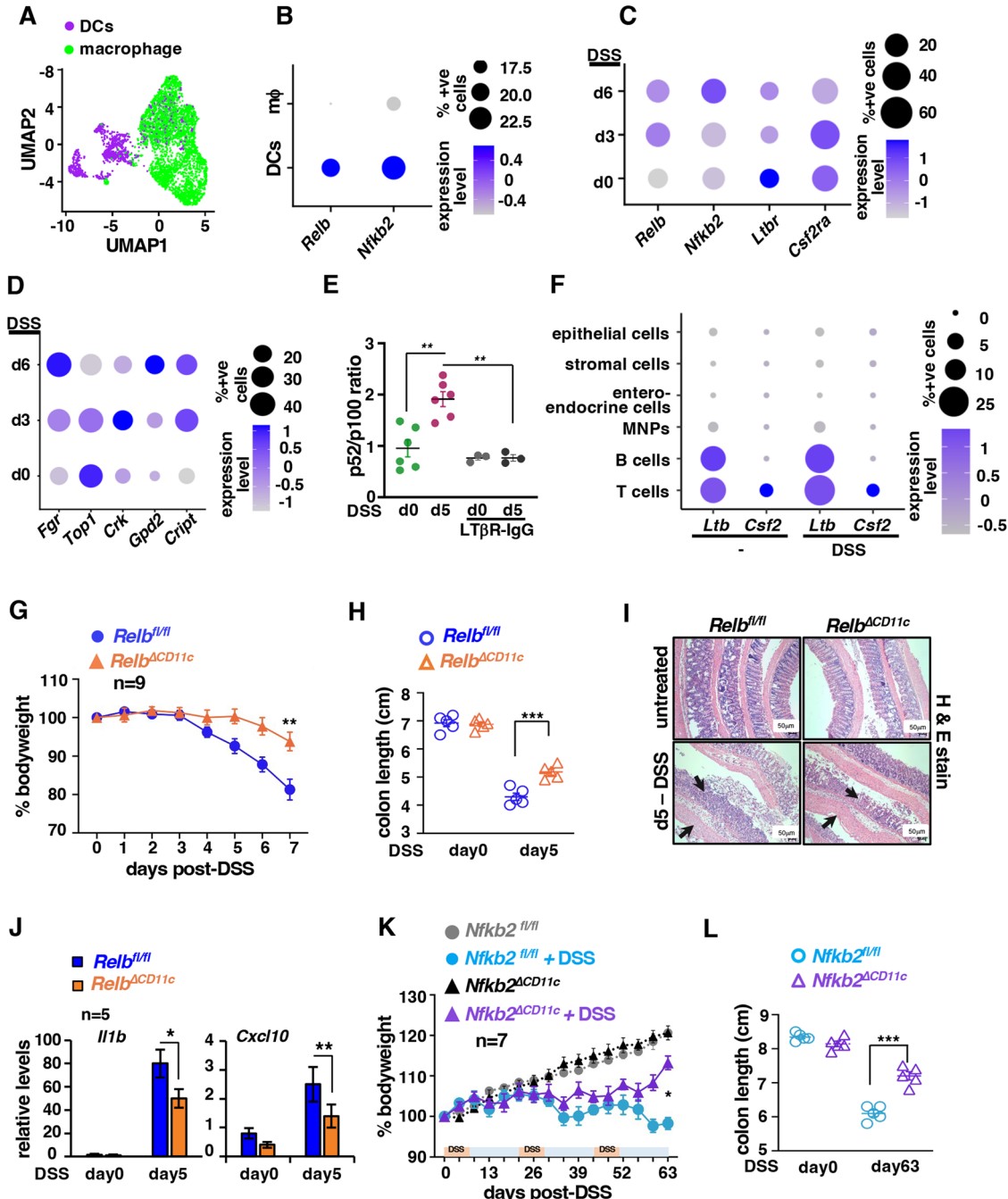

Next, we examined if the noncanonical NF-κB pathway in DCs caused colitogenic pathologies. To this end, we generated *Relb^ΔCD11c* mice by crossbreeding *Relb^fl/fl* mice with CD11c-Cre⁺ mice. As reported earlier (Andreas et al, 2019), we found efficient depletion of RelB in specifically splenic CD11c⁺ cells in *Relb^ΔCD11c* mice (Appendix Fig. S1F). We then subjected these mice to experimental colitis. As such, acute DSS challenge for 7 days led to up to 20% bodyweight loss in littermate *Relb^fl/fl* controls, accompanied by severe colon shortening and pervasive epithelial disruption associated with edema and leukocyte infiltration (Fig. 1G–I; Appendix Fig. S1G). *Relb^ΔCD11c* mice presented significantly less

reduction in bodyweight over the course of DSS treatment and a modest decrease in colon length with a relatively intact epithelium at day 5 (Fig. 1G–I). The intestinal epithelial architecture of untreated *Relb^fl/fl* and *Relb^ΔCD11c* mice was histologically indistinguishable. RelB deficiency in DCs also restricted the DSS-induced increase in barrier permeability in knockout mice (Appendix Fig. S1H). Similarly, our mRNA analyses revealed diminished DSS-induced expressions of inflammatory genes, including those encoding IL-1β and CXCL10, in the colon of *Relb^ΔCD11c* mice (Fig. 1J). Therefore, DC-intrinsic RelB deficiency moderated inflammation and acute colitis in mice.

**Figure 1.  Charting the role of the noncanonical NF-κB pathway in DCs in modulating experimental colitis in mice.**

(A) UMAP depicting the DC and macrophage cell population among mononuclear phagocytes in the mouse colon. Single-cell RNA-seq data available at the NCBI database was used for analyses (Data ref: Ho et al, 2021). (B) Dot plot comparing DCs and macrophages in the mouse colon for the expression of *Relb* and *Nfkb2* mRNAs. (C, D) Dot plot revealing quantified levels of indicated mRNAs in intestinal DCs from the WT mice left untreated or subjected to DSS treatment. (E) Boxplot showing the abundance of p52 in relation to p100 in intestinal DCs, as determined by quantifying the respective band intensities in immunoblot analyses. DCs were collected from MLNs of untreated mice or LTβR-IgG-treated mice, administered with 2% DSS in drinking water for 5 days. Each dot represents one biological sample (**$P = 0.0014$, **$P = 0.0011$). (F) Dot plot revealing quantified levels of indicated mRNAs in different cells from colon of WT mice left untreated or subjected to DSS treatment. Single-cell RNA-seq data available at the NCBI database was used for analyses (Data ref: Ho et al, 2021). (G, H) *Relb*$^{fl/fl}$ and *Relb*$^{ΔCD11c}$ mice were subjected to acute DSS treatment for 7 days and changes in the bodyweight ($n = 9$, **$P = 0.005$) (G) and colon length ($n = 5$, ***$P = 0.0009$) (H) were monitored. (I) Representative images of H&E-stained colon sections from the indicated mice left untreated or subjected to acute DSS treatment. The data represent $n = 3$; 4 fields per section and a total of three sections from each set were examined. Black arrows indicate edema, leucocyte infiltration and epithelial lining damage. Scale bar, 50 μm. (J) RT-qPCR demonstrating the colonic abundance of indicated mRNAs in untreated or DSS-treated *Relb*$^{fl/fl}$ and *Relb*$^{ΔCD11c}$ mice. ($n = 5$, *$P = 0.028$, **$P = 0.0032$) (K, L). In a chronic colitis regime, *Nfkb2*$^{fl/fl}$ and *Nfkb2*$^{ΔCD11c}$ mice were subjected to three cycles of DSS treatment, where each cycle involved 7 days of 1% DSS treatment followed by 14 days of recovery. Subsequently, changes in the bodyweight ($n = 7$, *$P = 0.010$) (K) and colon length ($n = 5$, ***$P = 0.0001$) (L) were measured. "$n$" represents number of biological replicates. Data represents mean ± SEM. For statistical analyses, two-tailed Student's $t$ test was performed. ns not significant. In our experiments involving knockouts, littermate male mice of the indicated genotypes were cohoused for at least 1 week prior to experiments. Source data are available online for this figure.

The coordinated functioning of proteins encoded by *Relb* and *Nfkb2* transduces noncanonical NF-κB signals. Therefore, we also generated *Nfkb2*$^{ΔDC}$ strains, which exhibited a lack of p100 expression in splenic CD11c$^+$ cells, but not in CD11c$^-$ or CD4$^+$ cells (Appendix Fig. S1I,1J). Compared to *Nfkb2*$^{fl/fl}$ controls, however, *Nfkb2*$^{ΔCD11c}$ mice displayed only subtly improved bodyweight phenotype upon acute DSS treatment (Appendix Fig. S1K,S1L). Interestingly, cyclical challenges with a low dose of DSS in the chronic chemical colitis regime resulted in a significantly less bodyweight reduction and colon shortening in *Nfkb2*$^{ΔCD11c}$ mice at the end of the third cycle on day 63 (Fig. 1K,L). It has been reported that an alternate RelB:p50 heterodimer functionally compensates, albeit partially, for the absence of RelB:p52 activity in *Nfkb2*-deficient cells (Basak et al, 2008). We reasoned that the molecular redundancies between RelB NF-κB factors obscured the phenotypic penetrance of *Nfkb2*$^{ΔCD11c}$ mice in the acute colitis regime and that repeated DSS exposure unmasked colitis-resilient phenotypes in these mice because of exaggerated DC-mediated immune reactions in the chronic regime (Wirtz et al, 2017). Overall, we infer that experimental colitis in mice strengthens in DC the noncanonical NF-κB pathway, which exacerbates inflammatory intestinal pathologies in mice.

## Noncanonical NF-κB signaling restrains tolerogenic Raldh2 activity in DCs

To mechanistically understand the immunomodulatory functions of noncanonical NF-κB signaling, we set out to examine bone marrow-derived dendritic cells (BMDCs) from WT or knockout mice (Appendix Fig. S2A). We followed a BMDC differentiation protocol that produced ~85% CD11c$^+$ BMDCs otherwise devoid of CD115$^{high}$ macrophage-like cells in the culture by day 9 (Jin and Sprent, 2018) (Appendix Fig. S2B,S2C). Notably, DC differentiation was associated with increased processing of p100 into p52 and elevated nuclear RelB and p52 levels (Fig. 2A). Our RNA-seq studies demonstrated that ablation of this constitutive noncanonical NF-κB signaling in *Nfkb2*$^{-/-}$ BMDC modified global gene expressions basally (Appendix Fig. S2D). RA metabolism emerged as one of the top-ranking DC-relevant biological pathways enriched among genes differentially expressed between WT and *Nfkb2*$^{-/-}$ BMDCs (Fig. 2B). Concordantly, our gene set enrichment analysis (GSEA) revealed a significant enrichment of RA targets among

genes whose basal expressions were augmented in *Nfkb2*$^{-/-}$ BMDCs, indicative of strengthening autocrine RA actions in knockouts (Fig. 2C). Sequential action of retinol dehydrogenases (Rdh) and retinal dehydrogenases (Raldh) convert retinol to RA, which is catabolized by the Cyp26 family of enzymes (Kedishvili, 2016) (Fig. 2D). Further probing our RNA-seq data for the expression of genes encoding these enzymes revealed a consistently elevated mRNA level of Raldh2 in *Nfkb2*$^{-/-}$ BMDCs (Fig. 2E). These studies indicated a possible role of noncanonical NF-κB signaling in transcriptionally modulating the RA pathway.

To substantiate noncanonical NF-κB-mediated control of the RA pathway, we derived BMDCs from *Relb*$^{ΔCD11c}$ or *Nfkb2*$^{ΔCD11c}$ mice. As expected, ex vivo differentiated BMDCs from *Relb*$^{ΔCD11c}$ or *Nfkb2*$^{ΔCD11c}$ mice produced only a minor amount of RelB or p100, respectively (Appendix Fig. S2E,S2F). In comparison to corresponding gene-sufficient floxed controls, *Relb*$^{ΔCD11c}$ and *Nkfb2*$^{ΔCD11c}$ BMDCs both displayed a ~ 4.5-fold increase in the basal expression of Raldh2 mRNA in our RT-qPCR analyses (Fig. 2F). LPS treatment did not discernably alter Raldh2 mRNA levels in WT or knockouts. In contrast, basal and LPS-induced IL-10 mRNA expressions were heightened in *Relb*$^{ΔCD11c}$, but not *Nfkb2*$^{ΔCD11c}$ BMDCs. Because of the shared colitis resilience phenotype of *Relb*$^{ΔCD11c}$ and *Nkfb2*$^{ΔCD11c}$ mice, we focused on Raldh2, whose expression was similarly affected upon ablation of either *Relb* or *Nfkb2* in BMDCs. We detected a 3.8-fold excess accumulation of Raldh2 protein in immunoblot analyses and a 2.2-fold increase in the basal Raldh enzymatic activity in the Aldefluor assay in *Relb*$^{ΔCD11c}$ BMDCs (Fig. 2G,H). *Nfkb2*$^{ΔCD11c}$ BMDCs revealed similar increases in the Raldh2 protein level and enzymatic activity (Appendix Fig. S2G,S2H). Previous studies involving DC-T-cell co-culture established RA as an important determinant of LP DC-mediated conversion of naive T cells into Tregs ex vivo (Sun et al, 2007). Our own analyses revealed that *Relb*$^{ΔCD11c}$ BMDCs were more proficient in converting naive splenic T cells into CD25$^+$Foxp3$^+$ CD4 Tregs ex vivo, and BMS493, a retinoic acid receptor (RAR) antagonist, erased this RelB-effect in DC-T-cell co-culture (Fig. 2I). Our flow cytometry analyses assured appropriate differentiation of bone marrow cells from *Relb*$^{ΔCD11c}$ mice into BMDCs in our culture (Appendix Fig. S2I). Taken together, DC differentiation ex vivo appears to trigger constitutive activation of noncanonical NF-κB signaling, which suppresses the tolerogenic Raldh2-RA axis.

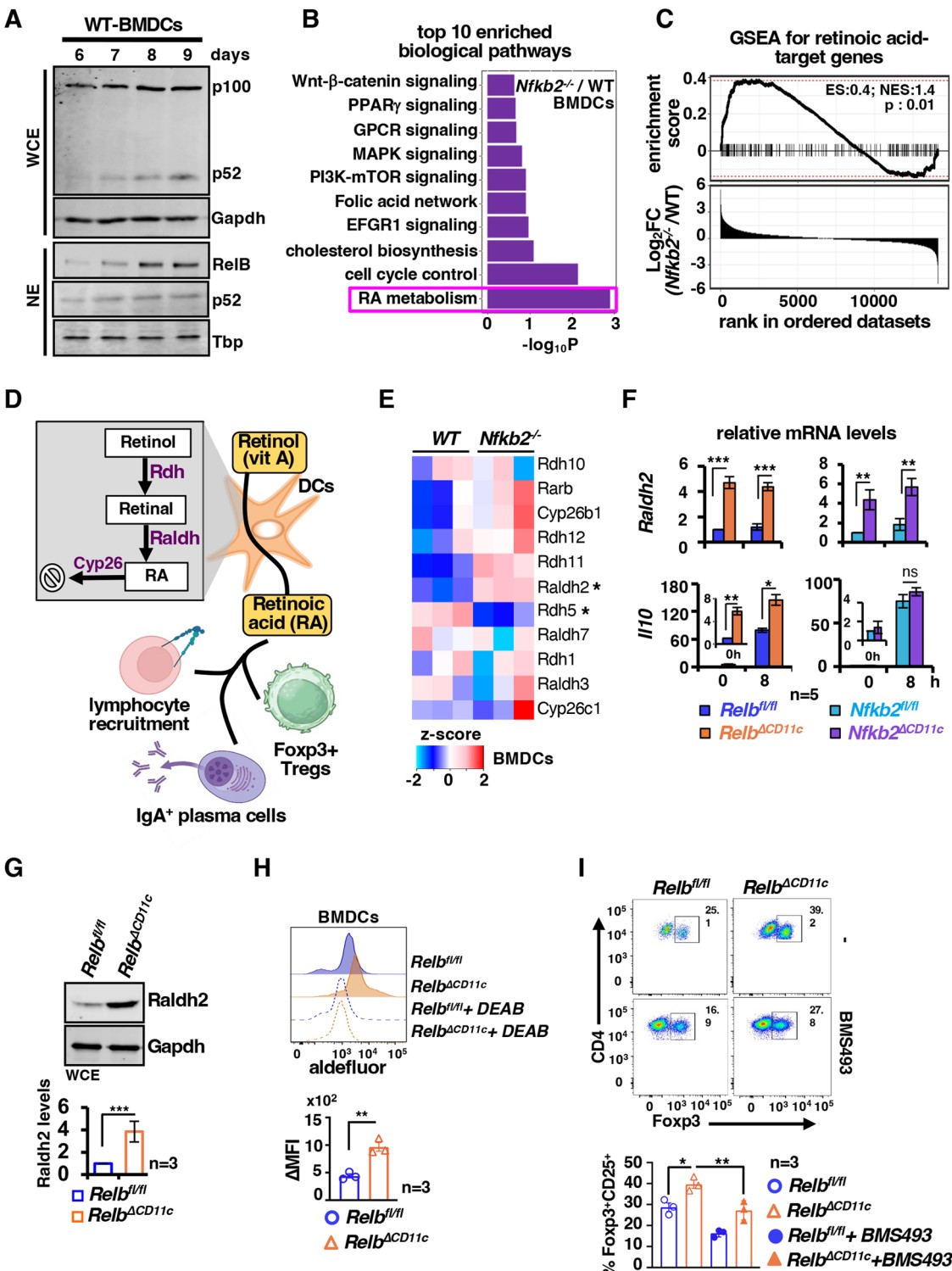

**DC functions of the noncanonical NF-κB signal transducers limit the frequency of FoxP3⁺ Treg in the mouse colon**

Next, we examined immunoregulatory attributes of noncanonical NF-κB-deficient DCs in vivo. Deletion of *Relb* or *Nfkb2* in CD11c⁺

cells did not significantly alter the frequency of DCs in MLNs or lamina propria (LP) in the colon (Fig. 3A; Appendix Fig. S3A). Also, when examined for the surface expression of CD80 and CD86 in flow cytometry analyses, intestinal DCs from *Relb^{fl/fl}* and *Relb^{ΔCD11c}* mice were indistinguishable (Fig. 3B). Similarly, *Nfkb2^{fl/fl}* and *Nfkb2^{ΔCD11c}* mice displayed equivalent levels of these co-stimulatory molecules on

◄

**Figure 2. Defective *Relb* or *Nfkb2* functions enhance the expression and activity of Raldh2 in DCs.**

(A) Immunoblot charting the abundance of p52 and p100 in whole-cell extracts (top) or RelB and p52 in nuclear extracts (bottom) in cells harvested from BMDC-differentiating culture at the indicated days. (B) Enrichment analyses revealing top ten enriched DC-relevant biological pathways among genes differentially expressed between $Nfkb2^{-/-}$ and WT BMDCs. For each genotype, RNA-seq data from three independent biological replicates were used. Benjamini–Hochberg method was used to calculate the P value. WikiPathways subset of cellular processes available at the Molecular Signatures Database was used for the enrichment analysis. (C) GSEA comparing $Nfkb2^{-/-}$ and WT BMDCs for the enrichment of RA targets among differentially expressed genes (top). Fold change (FC) in the basal mRNA level between these genotypes was considered to create a ranked gene list, which was subjected to GSEA using a previously published list of RA-target genes (see "Methods" for details). Benjamini–Hochberg method was used to calculate the P value. Vertical lines represent individual RA targets. ES and NES denote enrichment and normalized enrichment scores, respectively. (D) A cartoon describing the RA metabolism pathway and its functions in DCs. (E) Heatmap showing the expression of selected RA metabolism-associated genes in WT and $Nfkb2^{-/-}$ BMDCs, as determined in RNA-seq. (F) RT-qPCR comparing untreated or LPS-treated BMDCs derived from $Relb^{fl/fl}$ and $Relb^{\Delta CD11c}$ mice (left, ***P = 0.0007, 0.0001, **P = 0.003, *P = 0.012) or $Nfkb2^{fl/fl}$ and $Nfkb2^{\Delta CD11c}$ mice (right, **P = 0.0012, 0.002, ns=0.33, P = 0.16) for gene expressions (n = 5). (G) Representative immunoblot revealing the abundance of Raldh2 in $Relb^{fl/fl}$ and $Relb^{\Delta CD11c}$ BMDCs. Corresponding band intensities were quantified and presented as a barplot below (n = 3, ***P = 0.001). (H) Histograms showing the Raldh enzymatic activity measured by Aldefluor assay in BMDCs. The Raldh inhibitor DEAB was used as a control. For a given genotype, ΔMFI was calculated as = $MFI_{no\ DEAB}$ - $MFI_{DEAB}$, and presented in a barplot (n = 3, **P = 0.0037). (I) Flow cytometry analyses demonstrating the generation of Foxp3+ CD4 Treg in DC-T-cell co-culture ex vivo. Briefly, BMDCs were cultured with naive T cells, isolated from the spleen of WT mice, in the presence of CD3 and TGF-β for 3.5 days. In specified sets, the RAR inhibitor BMS493 was added. The frequency of Foxp3+ Treg as a percentage of CD4 cells was determined and presented in the barplot (bottom) (n = 3, *P = 0.017, **P = 0.005). "n" represents the number of biological replicates. Data represent mean ± SEM. Two-tailed Student's t test was performed. ns-not significant. Source data are available online for this figure.

intestinal DCs (Appendix Fig. S3B). Also, our analyses revealed only a minor alteration in the relative frequency of intestinal DC subsets in $Relb^{\Delta CD11c}$ mice (Appendix Fig. S3C). Corroborating BMDCs studies, intestinal DCs from $Relb^{\Delta CD11c}$ mice, but not from $Nfkb2^{\Delta CD11c}$ mice, expressed significantly more IL-10 (Fig. 3B,C). However, RelB-dependent IL-10 regulation was more apparent in MLN DCs. Importantly, we could confirm an increased level of Raldh2 mRNA in intestinal DCs from both $Relb^{\Delta CD11c}$ and $Nfkb2^{\Delta CD11c}$ mice (Fig. 3D). Our enzymatic assay substantiated a more than 2.1-fold rise in the Raldh activity in intestinal DCs from untreated $Relb^{\Delta CD11c}$ mice (Fig. 3E). Thus, noncanonical NF-κB signaling regulated Raldh2 activity in DCs in vivo. Raldh2 in intestinal DCs drives the synthesis of RA, which supports FoxP3 expressions and upregulates gut-homing receptors α4β7 and CCR9 in Tregs (Iwata et al, 2004; Lu et al, 2014). As such, inactivating *Relb* or *Nfkb2* in CD11c+ cells did not discernably alter the frequency of total CD4 T lymphocytes in the colon of untreated or DSS-treated mice (Appendix Fig. S3D). However, the FoxP3+ Treg frequency among CD4 T cells in LP and MLNs indeed improved from 18.7 and 7.9% in $Relb^{fl/fl}$ mice to 36.6% and 12.4% in $Relb^{\Delta CD11c}$ mice, respectively (Fig. 3F). While acute DSS treatment subtly skewed this Treg compartment at day 5, $Relb^{\Delta CD11c}$ mice maintained an overall increased level of these cells even in the colitogenic gut. Except for MLNs from untreated sets, the frequency of FoxP3+ Tregs was similarly augmented in the intestine of $Nfkb2^{\Delta CD11c}$ mice (Fig. 3G; Appendix Fig. S3E). Accordingly, we also observed a proportionate increase in the total Treg numbers in the mouse colon upon ablation of the noncanonical NF-κB pathway in DCs (Appendix Fig. S3F). Furthermore, we captured a reduction in the frequency of intestinal RORγt+ Th17 cells in our knockouts, presumably secondary to the expansion of the Treg compartment (Appendix Fig. S3G).

Nevertheless, FoxP3+ Tregs from MLNs of $Relb^{\Delta CD11c}$ mice were, if anything, subtly more efficient than those from $Relb^{fl/fl}$ counterparts in expressing IL-10 and α4β7, and displayed significantly elevated levels of CCR9 on their surface (Fig. 3H). Collectively, noncanonical NF-κB signaling impedes the Raldh2-mediated RA synthesis pathway in mucosal DCs, limiting the abundance of Tregs and their gut-homing receptor expressions in the intestine.

## DC-intrinsic RelB function controls the abundance of IgA-secreting cells in the colon and gut microbiome

Previous studies suggested that RA promotes IgA-secreting cells at the gut mucosal interface (Villablanca et al, 2011). We further explored whether reinforcing the RA pathway in noncanonical NF-κB-deficient DCs impacted intestinal sIgA levels. Our flow cytometry analyses disclosed a ~2.5-fold increase in the frequency of IgA+B220+ cells in LP of $Relb^{\Delta CD11c}$ mice (Fig. 4A,B), and our ELISA ascertained a significantly elevated fecal sIgA levels in this knockout (Fig. 4C). Because sIgA coating is known to shape the gut microbiome (Nakajima et al, 2018; Pabst and Izcue, 2022), we performed 16 s rRNA gene sequencing of fecal DNA to estimate the prevalence of different microbial taxa. We found that RelB deficiency was associated with changes specifically in two phyla; it led to an increase in the abundance of Firmicutes and a reduction in the level of Actinobacteriota (Fig. 4D). Another prominent phylum Bacteroidota showed a comparable fecal abundance among $Relb^{fl/fl}$ and $Relb^{\Delta CD11c}$ mice. Genus-level analyses of sequencing data similarly revealed gut microbiome alterations in these mice, including an expansion of *Sangeribacter* and *Lactobacillus* but also a reduction of *Akkermansia* and *Faecalibacterium* abundance (Fig. 4E). Corroborating DNA sequencing studies, our qPCR analyses of fecal DNA substantiated changes in Firmicutes and Actinobacteriota in $Relb^{\Delta CD11c}$ mice (Fig. 4F). Moreover, MACS based fractionation of fecal pellet identified enrichment of *Enterobacter*, *Bifidobacterium*, Segmented filamentous bacteria (SFB), *Sutterella* and *Prevotella* among IgA-bound gut microbes in $Relb^{\Delta CD11c}$ mice (Fig. 4G). We also observed a decrease in the abundance of *β-proteobacteria* in the fecal IgA+ fraction in these knockouts.

Notably, Firmicutes, and more so the increased ratio of Firmicutes to Bacteroidota, have been linked to improved intestinal barrier integrity (Stojanov et al, 2020; Sun et al, 2022). Likewise, *Sangeribacter*, *Lactobacillus*, and *Faecalibacterium* are thought to largely promote intestinal health (Forster et al, 2022). Therefore, barring a reduction in *Faecalibacterium*, $Relb^{\Delta CD11c}$ mice had an overall preponderance of gut-beneficial taxa. More so, these changes were associated with differential sIgA coating of microbial entities. Importantly, differential sIgA coating not only quantitatively impacts the microbiome composition but also qualitatively

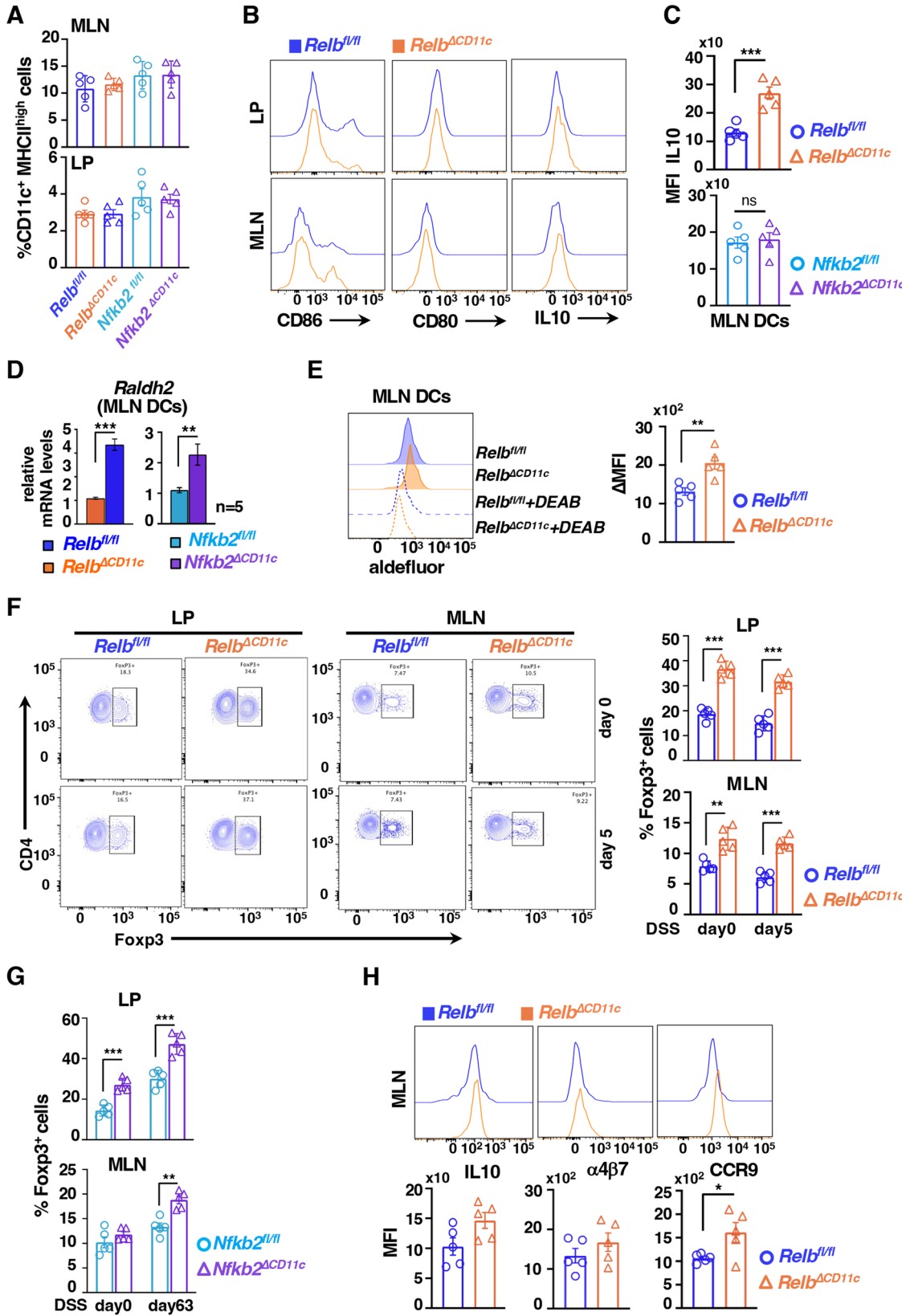

**Figure 3. Disruption of noncanonical NF-κB signaling in CD11c⁺ cells accumulate FoxP3⁺ Tregs in the mouse colon.**

(A) Flow cytometry studies indicating the abundance of DCs, as a percentage of CD45+ cells, in MLN or LP. ($n = 5$, "$n$" represents number of biological replicates). (B) Representative histograms showing the expression of CD80, CD86 and IL-10 in DCs from LP or MLN from $Relb^{fl/fl}$ and $Relb^{\Delta CD11c}$ mice. (C) Barplot revealing MFI for IL-10 expression in MLN DCs from the indicated mice ($n = 5$, ***$P = 0.0005$). (D) RT-qPCR depicting the abundance of Raldh2 mRNA in MLN DCs from $Relb^{fl/fl}$ and $Relb^{\Delta CD11c}$ or $Nfkb2^{fl/fl}$ and $Nfkb2^{\Delta CD11c}$ mice ($n = 5$, ***$P = 9.91 \times 10^{-5}$, **$P = 0.0056$). (E) Histograms capturing Raldh enzymatic activity in MLN DCs from $Relb^{fl/fl}$ and $Relb^{\Delta CD11c}$ mice. Quantified average MFI has been presented in a barplot. ($n = 5$, **$P = 0.0039$). (F) Representative contours from flow cytometry analyses showing the frequency of Foxp3⁺ Tregs as a percentage of CD4⁺ T cells in LP or MLNs of untreated or DSS-treated $Relb^{fl/fl}$ and $Relb^{\Delta CD11c}$ mice. Corresponding quantified data has been presented as a barplot in the right ($n = 5$, LP ***$P = 6.95 \times 10^{-6}$, $1 \times 10^{-5}$, MLN **$P = 0.0012$, ***$P = 1.59 \times 10^{-5}$). (G) Barplot revealing the mean frequency of Foxp3⁺ Treg in LP and MLN of $Nfkb2^{fl/fl}$ and $Nfkb2^{\Delta CD11c}$ mice ($n = 5$, LP ***$P = 0.00013$, $0.0004$, MLN **$P = 0.0011$). (H) Overlapping histograms depicting the expression of IL-10, α4β7, and CCR9 in Foxp3⁺ Tregs from MLNs of $Relb^{fl/fl}$ and $Relb^{\Delta CD11c}$ mice. Quantified average MFIs are indicated in the individual barplot below ($n = 5$, *$P = 0.036$). Each datapoint signifies an individual mouse; data represent mean ± SEM. Two-tailed Student's $t$ test was performed. ns not significant. Source data are available online for this figure.

affects gene expressions and metabolic functions of microbial entities in the intestinal niche (Nakajima et al, 2018). Curiously, $Nfkb2^{fl/fl}$ and $Nfkb2^{\Delta CD11c}$ mice were not discernibly different with respect to the frequency of intestinal IgA⁺B220⁺ cells, luminal sIgA levels and the abundance of Firmicutes and Actinobacteriota (Fig. 4B,C,F). Taken together, we deduce that a DC-intrinsic RelB function, where $Nfkb2$ is dispensable, limits the accumulation of intestinal IgA⁺ B cells and luminal IgA, which likely foster a gut microbiome protective against experimental colitis in mice.

## Noncanonical NF-κB signaling limits β-catenin-driven Raldh2 synthesis in DCs by directing Axin1 transcription

It was earlier reported that Notch2, Irf4, and β-catenin as key factors directing the transcription of $Raldh2$ in DCs (Manicassamy et al, 2010; Yashiro et al, 2018; Zaman et al, 2017). We found that RelB depletion increased the abundance of specifically β-catenin, and not Notch2 or Irf4, in BMDCs (Fig. 5A). Compared to respective controls, total β-catenin levels were elevated 4.2-fold in $Relb^{\Delta CD11c}$ and 3.6-fold in $Nfkb2^{\Delta CD11c}$ BMDCs (Fig. 5B). This was accompanied by a proportionate increase in the level of active β-catenin and nuclear β-catenin in these knockouts. Importantly, pharmacological inhibition of β-catenin-mediated transcription in noncanonical NF-κB-deficient BMDCs using iCRT3 restored the Raldh2 protein abundance to that observed in control cells (Fig. 5C). Therefore, our data supported a role of β-catenin in boosting Raldh2 production in the absence of noncanonical NF-κB signaling.

As such, Axin proteins tether β-catenin to a multiprotein destruction complex consisting of adenomatous polyposis coli (APC) and glycogen synthase kinase-3 (GSK-3) that promotes phosphorylation-dependent proteasomal degradation of β-catenin in resting cells (MacDonald et al, 2009) (Appendix Fig. S4A). Wnt ligands signal through Frizzled receptors to rescue -catenin from this constitutive degradation. We argued that strengthening autocrine signaling owing to increased Wnt secretion augmented β-catenin levels in knockouts. However, blocking the autocrine pathway with C59, a small molecule inhibitor of Wnt secretion from cells, did not discernibly change the abundance of β-catenin in $Relb^{\Delta CD11c}$ BMDCs (Fig. 5D). We then investigated if noncanonical NF-κB deficiency instead impacted other β-catenin regulators. Examining our RNA-seq data indeed identified a significant decline in the mRNA level of the Axin1 isoform in $Nfkb2^{-/-}$ BMDCs (Fig. 5E). When we analyzed the published RelB ChIP-seq dataset generated using WT BMDCs, the genome browser track of $Axin1$ revealed two distinct RelB peaks associated with a promoter and an intronic enhancer-κB element (Fig. 5F). Our RT-qPCR studies substantiated that a dysfunctional non-canonical NF-κB pathway caused elevated expressions of Axin1 mRNA and protein in $Relb^{\Delta CD11c}$ and $Nfkb2^{\Delta CD11c}$ BMDCs (Fig. 5G,H). Notably, our immunoprecipitation experiments demonstrated that depleting Axin1 levels in $Relb^{\Delta CD11c}$ BMDCs impeded the association of β-catenin with GSK-3β, a critical component of the destruction complex (Fig. 5I). We conclude that noncanonical NF-κB-mediated transcriptional control of Axin1 suppresses β-catenin-driven Raldh2 expressions in DCs.

## β-catenin determines immunoregulatory DC functions of the noncanonical NF-κB pathway in the mouse gut

Because DC-intrinsic β-catenin functions were shown to limit mucosal inflammation in the intestine (Manicassamy et al, 2010; Zhao et al, 2018), we asked if β-catenin accumulated in noncanonical NF-κB-deficient DCs was responsible for the resilience of these knockout mice to colitogenic insults. Corroborating our ex vivo analyses, we observed a approximately threefold increase in the abundance of β-catenin in intestinal DCs from $Relb^{\Delta CD11c}$ or $Nfkb2^{\Delta CD11c}$ mice compared to those from respective control floxed mice (Fig. 6A; Appendix Fig. S4B). These changes were associated with a substantial more than two-fold increase in the frequency of β-catenin^high cells among MLN DCs in $Relb^{\Delta CD11c}$ mice. To causally establish the role of β-catenin in noncanonical NF-κB-deficient DCs, we then introduced a DC-specific β-catenin haploinsufficiency in mice devoid of RelB expression in DCs. Convincingly, MLN DCs from $Ctnnb1$ ^hetCD11c$Relb^{\Delta CD11c}$ mice displayed a significant drop in the Raldh2 activity from those observed in $Relb^{\Delta CD11c}$ mice (Fig. 6B). Consistent with the tolerogenic role of DC-expressed Raldh2, the frequency of colonic FoxP3⁺ Tregs and fecal sIgA in $Ctnnb1$ ^hetCD11c$Relb^{\Delta CD11c}$ mice were restored to those found in $Relb^{fl/fl}$ mice (Fig. 6C,D). When challenged in the acute regime, $Ctnnb1$ ^hetCD11c$Relb^{\Delta CD11c}$ mice exhibited bodyweight loss and colon shortening at day 6, which were rather analogous to $Relb^{fl/fl}$ mice (Fig. 6E,F). These studies illustrate a noncanonical NF-κB-driven DC network that aggravates intestinal inflammation by subverting the tolerogenic β-catenin–Raldh2-RA axis.

## Human IBD is associated with strengthening noncanonical NF-κB signaling in intestinal DCs

Finally, we sought to examine the relevance of our newly identified DC network in human IBD. To this end, we analyzed single-cell

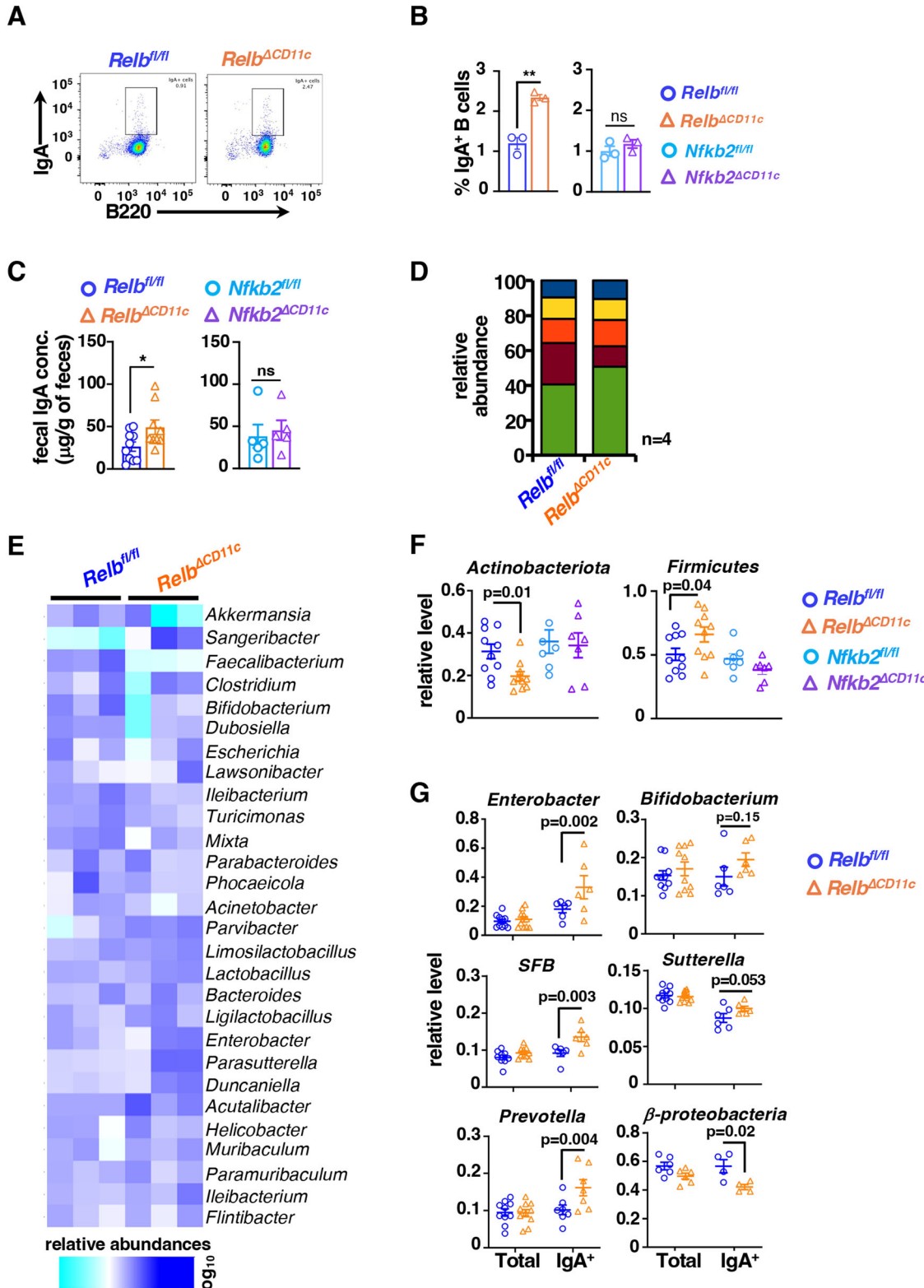

◄

**Figure 4. RelB NF-κB deficiency in CD11c⁺ cells as a determinant of the intestinal abundance of IgA-secreting cells and gut microbiome.**

(A) Representative FACS plots revealing the frequency of IgA⁺B220⁺ cells as a percentage of CD19⁺ cells in LP of $Relb^{fl/fl}$ and $Relb^{\Delta CD11c}$ mice. (B) Barplot demonstrating quantified frequency of IgA⁺ B220⁺ cells in LP of mice of the indicated genotypes ($n = 3$, **$P = 0.0016$, ns $P = 0.32$). (C) ELISA comparing $Relb^{fl/fl}$ and $Relb^{\Delta CD11c}$ mice for sIgA concentrations in fecal pellets. Each dot represents one biological sample (*$P = 0.04$, ns $= 0.75$). (D, E) The relative abundance of various microbial phyla ($n = 4$) (D) and genera ($n = 3$) (E) charted in barplot and heatmap, respectively. Briefly, 16 s rDNA sequencing was performed using fecal DNA from $Relb^{fl/fl}$ and $Relb^{\Delta CD11c}$ mice and prevalent phyla and genera were plotted. (F) 16 s rDNA-based qPCR analyses of fecal DNA showing the level of indicated microbial taxa in $Relb^{fl/fl}$ and $Relb^{\Delta CD11c}$ or $Nfkb2^{fl/fl}$ and $Nfkb2^{\Delta CD11c}$ mice. Each dot represents one biological sample. (G) qPCR analyses MACS-sorted IgA⁺ fractions of fecal pellets comparing $Relb^{fl/fl}$ and $Relb^{\Delta CD11c}$ mice for the prevalence of the indicated bacteria. Each dot represents one biological sample. Each datapoint represents an individual mouse; data represent mean ± SEM. Two-tailed Student's $t$ test was performed for (A–C) and two-way ANOVA was carried out for (F, G). ns not significant. Source data are available online for this figure.

RNA-seq data (Fig. 7A) generated using colon biopsies from 18 IBD patients and 12 healthy individuals (Smillie et al, 2019; Data ref: Smillie et al, 2019). Consistent with our observation involving the mouse colon, we found that among various MNPs, *RELB* and *NFKB2* mRNAs were mostly expressed in DCs (Fig. 7B). Compared to healthy controls, intestinal DCs in the inflamed tissue of IBD patients possessed a substantially elevated level of these mRNAs (Fig. 7C; Appendix Fig. S5A). Although less apparent, DCs from the non-inflamed tissue of IBD patients also revealed an increase in the abundance of these mRNAs. Importantly, we also noticed an almost equivalent expression of LTβR mRNA in these human DCs (Fig. 7C). We then determined the expression of RelB-important genes, viz *Fgr*, *Top1*, and *Crk*, in intestinal DCs from human subjects. Examining single-cell data revealed augmented DC levels of *FGR*, *TOP1*, *CRK* and *CRIPT* mRNAs in the inflamed as well as non-inflamed tissues from IBD patients that were associated with an increased frequency of DCs expressing these mRNAs in colonic tissues (Fig. 7D). To further substantiate, we examined an additional single-cell RNA-seq dataset from IBD patients (Devlin et al, 2021; Data ref: Devlin et al, 2021). Interrogation of this dataset confirmed that *RELB* and *NFKB2* mRNAs were majorly expressed in intestinal DCs as opposed to macrophages and that IBD patients displayed elevated expressions of RelB-important genes in DCs in comparison to healthy individuals (Appendix Fig. S5B–S5D). Therefore, our analyses suggested that human IBD was associated with heightened noncanonical NF-κB signaling in intestinal DCs.

Next, we probed if, corroborating our observation in knockout mice, strengthening noncanonical NF-κB signaling in DCs weakened the β-catenin–Raldh2-RA axis in IBD patients. To this end, we examined single-cell RNA-seq data from IBD patients (Smillie et al, 2019) for the expression of known β-catenin-regulated genes as a proxy for β-catenin's transcriptional activity. Indeed, we could demonstrate that *CCND1*, a well-recognized β-catenin-activated gene, was expressed by ~20% of intestinal DCs and substantially downmodulated in DCs located particularly in the inflamed tissues of IBD patients (Fig. 7E). We then subjected this dataset to GSEA. Corroborating previous studies linking IBD to decreased Raldh activity in intestinal DCs (Magnusson et al, 2016), our analyses revealed a significant enrichment of RA targets among genes that were downmodulated in DCs derived from inflamed colonic tissues of IBD patients as compared to those from non-inflamed tissues (Fig. 7F). These studies substantiated the inverse correlation between noncanonical NF-κB signaling and the RA pathway in DCs in the inflamed human gut.

Furthermore, IBD was associated with a substantial decrease in the frequency of IL-10-expressing Tregs (Fig. 7G, left), as observed earlier (Liu et al, 2012). We also noticed a reduction in the average

expression of *IGHA1* and *IGHA2*, genes encoding IgA heavy chain, in colonic plasma cells of the IBD patients compared to the non-IBD subjects (Fig. 7G, right). Collectively, our investigation argues that the noncanonical NF-κB pathway targets β-catenin to limit the tolerogenic Raldh2 functions in intestinal DCs, orchestrating local inflammation in colitogenic mice as well as in IBD patients (Fig. 8).

## Discussion

Despite the known immunoregulatory role of the noncanonical NF-κB pathway in DCs (Hofmann et al, 2011; Huang et al, 2018; Jie et al, 2018; Shih et al, 2012), whether RelB:p52 distorts DC functions in intestinal pathologies remains unclear.

Our investigation identified intensified noncanonical NF-κB signaling in intestinal DCs from colitogenic mice and causally linked this pathway to exacerbated experimental colitis (Fig. 1). DC-derived RA imparts tolerogenic attributes, and β-catenin induces the expression of the RA-synthesizing enzyme Raldh2 in intestinal DCs (Manicassamy et al, 2010; Iwata et al, 2004; Bos et al, 2022; Coombes et al, 2007). Our mechanistic studies suggested that RelB:p52 transcriptionally supported Axin1, which restrained Raldh2 synthesis in DCs by downmodulating β-catenin (Figs. 2 and 5). Concordantly, noncanonical NF-κB impairment reinforced β-catenin-mediated Raldh2 expressions in DCs, improving the colonic frequency of protective Tregs and IgA⁺ B cells and also luminal sIgA levels, which were associated with a modified gut microbiome (Figs. 3 and 4). Indeed, introducing β-catenin haploinsufficiency in RelB-deficient DCs reinstated the colitogenic sensitivity in mice (Fig. 6). Corroborating our studies involving knockout mouse strains, the inflamed colon from IBD patients harbored DCs displaying heightened noncanonical NF-κB signaling and reduced Raldh activity, presumable owing to diminished β-catenin functions (Fig. 7). Consistently, IBD patients also revealed a diminished frequency of FoxP3⁺ Tregs and IgA-secreting cells in the intestinal niche. IBD is multifactorial, with the involvement of several environmental, genetic, and immune components. Our analyses led us to propose DC functions of the noncanonical NF-κB pathway as a contributing factor to intestinal pathologies. In this mechanistic model, noncanonical NF-κB signaling subverts β-catenin-dependent, Raldh2-mediated synthesis of RA, which enforces a tolerogenic environment in the inflamed gut by promoting Tregs and luminal IgA (Fig. 8). Previous studies suggested that LTβR modulates intestinal DC functions (Tumanov et al, 2011). Our analyses further placed the noncanonical NF-κB pathway downstream of LTβR in these DCs. However, further studies will be important to elucidate additional triggers of this

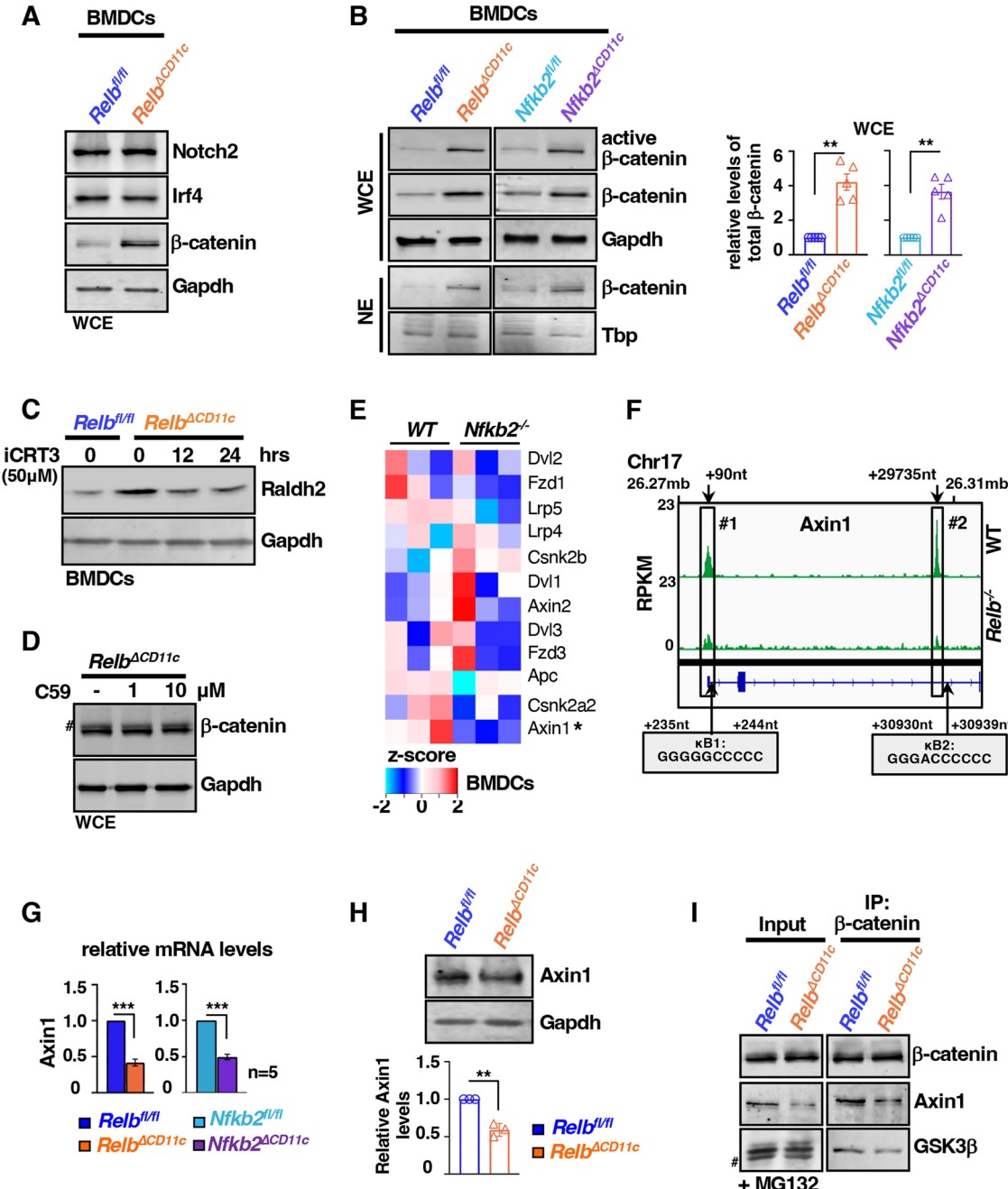

**Figure 5. Reduced transcription of Axin1 promotes β-catenin-driven Raldh2 synthesis in noncanonical NF-κB-deficient DCs.**

(A) Immunoblot analyses comparing *Relb^fl/fl* and *Relb^ΔCD11c* BMDCs for the abundance of the indicated transcription factors. (B) Representative immunoblot revealing active and total β-catenin levels in whole-cell or nuclear extracts derived from BMDCs of the indicated genotypes. Total β-catenin band intensities were also quantified and presented as barplot (right). Gapdh and Tbp served as loading controls for whole-cell and nuclear extracts, respectively ($n = 5$, **$P = 0.0035$, 0.0016, Welch's $t$ test). (C) Immunoblot charting Raldh2 mRNA level in *Relb^fl/fl* and *Relb^ΔCD11c* BMDCs left untreated or treated with 50 μM of iCRT3, which inhibits β-catenin-mediated transcription, for 12 h. (D) Immunoblot analyses capturing the abundance of β-catenin in *Relb^ΔCD11c* BMDCs treated for 24 h with 10 μM of C59, which prevents secretion of Wnt ligands by inhibiting Porcupine. (E) Heatmap comparing the abundance of mRNAs encoding the indicated β-catenin regulators in WT and *Nfkb2^-/-* BMDCs, as determined by RNA-seq. (F) The genome browser track of RelB ChIP-seq peaks in WT BMDCs (Data ref: Navarro et al, 2023) for *Axin1*. Cognate κB motifs associated with these peaks have also been indicated. (G, H) RT-qPCR and immunoblot analyses scoring the level of Axin1 mRNA ($n = 5$, ***$P = 3.45 \times 10^{-9}$, $1 \times 10^{-6}$) (G) and protein ($n = 3$, **$P = 0.006$) (H), respectively, in BMDCs. For immunoblot analyses, corresponding band intensities were quantified and presented (bottom). (I) Immunoblot analyses of β-catenin co-immunoprecipitants obtained using *Relb^fl/fl* and *Relb^ΔCD11c* BMDCs subjected to MG-132 treatment for 4 h. Left, MG-132-treated cell extracts used as inputs for immunoprecipitation. MG-132 treatment was carried out to achieve comparable β-catenin levels between WT and knockout BMDCs. # represents non-specific bands. "$n$" represents the number of biological replicates. Quantified data represent the mean of at least three experimental replicates ± SEM. Two-tailed Student's $t$ test was performed. Source data are available online for this figure.

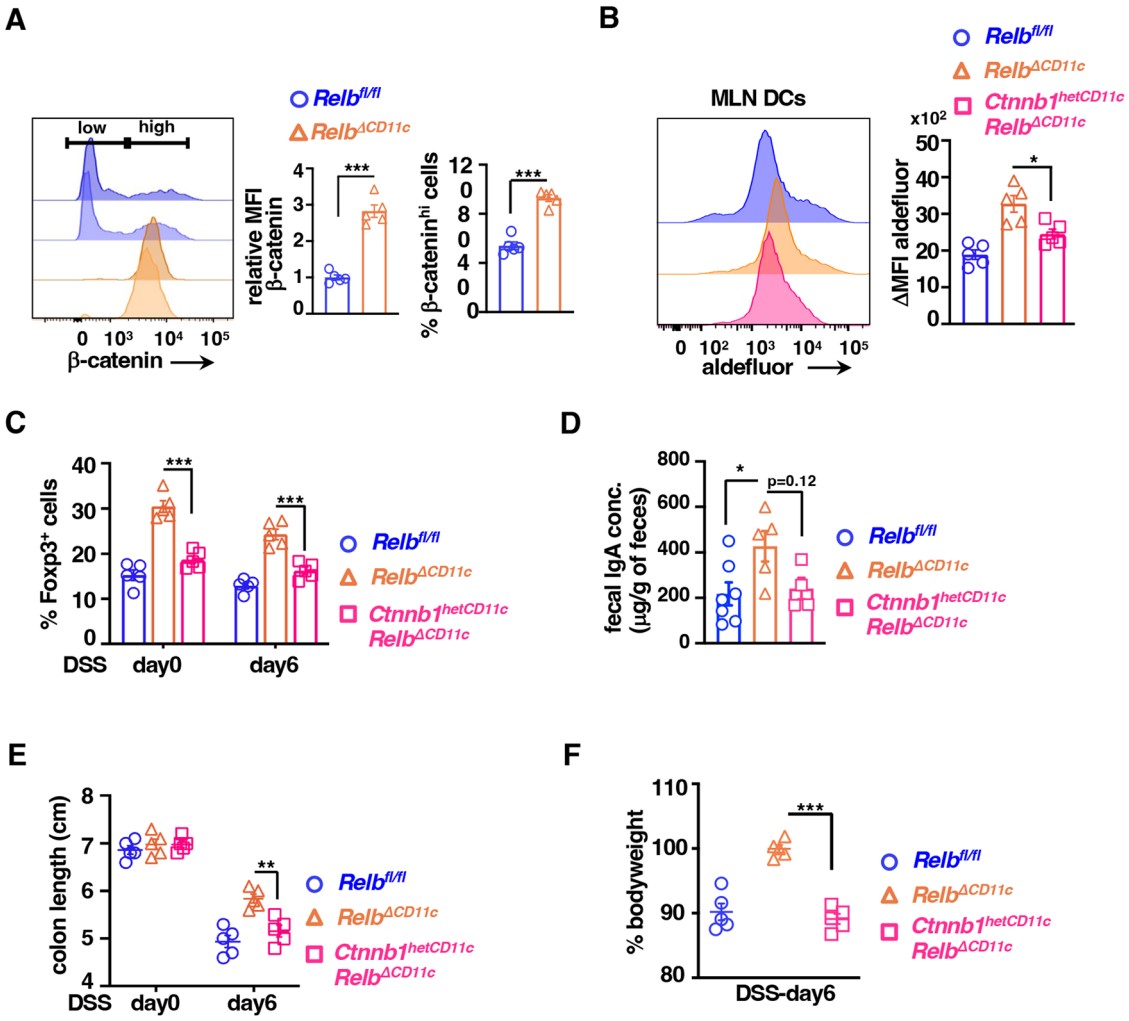

**Figure 6. Haploinsufficiency of β-catenin in noncanonical NF-κB-deficient DCs reinstates the colitogenic phenotype in mice.**

(A) Histograms comparing *Relb^fl/fl* and *Relb^ΔCD11c* mice for the expression of β-catenin in LP DCs. Barplot shows quantified MFI for β-catenin expression and also the frequency of β-catenin high cells as a percentage of LP DCs (n = 5, ***P = 0.0001, ***P = 0.0001, unpaired *t* test). (B) Histograms revealing the Raldh activity in MLN DCs from *Relb^fl/fl*, *Relb^ΔCD11c*, and *Ctnnb1^hetCD11c Relb^ΔCD11c* mice. Corresponding ΔMFI values calculated from corresponding DEAB controls have been presented in the barplot (n = 5, *P = 0.014). (C) Barplot revealing the mean frequency of Foxp3+ Tregs in LP of indicated mice (n = 5, ***P = 0.0002, 0.0008). (D) ELISA comparing fecal IgA levels of the indicated genotypes. Each dot represents one biological sample (*P = 0.043, ns = 0.125, unpaired *t* test). (E, F) Scatter plots revealing colon length (E) and bodyweight changes (F) on day 6 upon acute DSS treatment of indicated mice (n = 5, **P = 0.0042, ***P = 0.0001). Each datapoint represents an individual mouse; data represent mean ± SEM. For statistical analysis, one-way ANOVA was performed, unless otherwise mentioned. ns not significant. Source data are available online for this figure.

pathway in DCs and to examine protective and deleterious LTβR functions in other intestinal cell subsets.

It was earlier reported that DC-specific ablation of *Relb*, but not *Nfkb2*, led to a systemic increase in the frequency of FoxP3+ Tregs (Andreas et al, 2019). We found a convergence of *Relb^ΔCD11c* and *Nfkb2^ΔCD11c* mice with respect to immune phenotypes—they both accumulated FoxP3+ Tregs in the intestine and were resilient to experimental colitis. Our data contended that systemic Treg effects were less important and that β-catenin-dependent modulation of specifically the intestinal Treg compartment was critical for the colitogenic role of RelB:p52 in DCs. As opposed to a substantial increase in the intestinal Treg frequency in *Relb^ΔCD11c* mice, RelB-depleted DCs ex vivo displayed only modest, albeit RA pathway-dependent, enhancement of Treg generation from naive T cells. Based on the literature (Lu et al, 2014) and our study, we reason that

in addition to its role in priming Treg differentiation, DC-derived RA also promoted gut homing of Tregs and their stabilization in the intestinal niche in the knockout mice. Additionally, RelB in DCs was shown to promote microbiota-reactive RORγt+ Tregs in the small intestine (Andreas et al, 2019). Although our study did not distinguish between tissue- and microbiota-reactive Tregs in the colon, tissue Tregs were also found to be critical for tolerance to gut commensals (Cebula et al, 2013). Of note, it was suggested that NIK in DCs induced the expression of IL-23, which strengthened Th17 responses to *Citrobacter rodentium* (Jie et al, 2018). A similar DC-intrinsic role of RelB in promoting IL-23-mediated Th17 response to gut commensals cannot be ruled out. However, our ex vivo and in vivo studies argues that Th17 regulations by RelB in DC could be secondary noncanonical NF-κB-mediated Treg controls, at least in the intestinal niche.

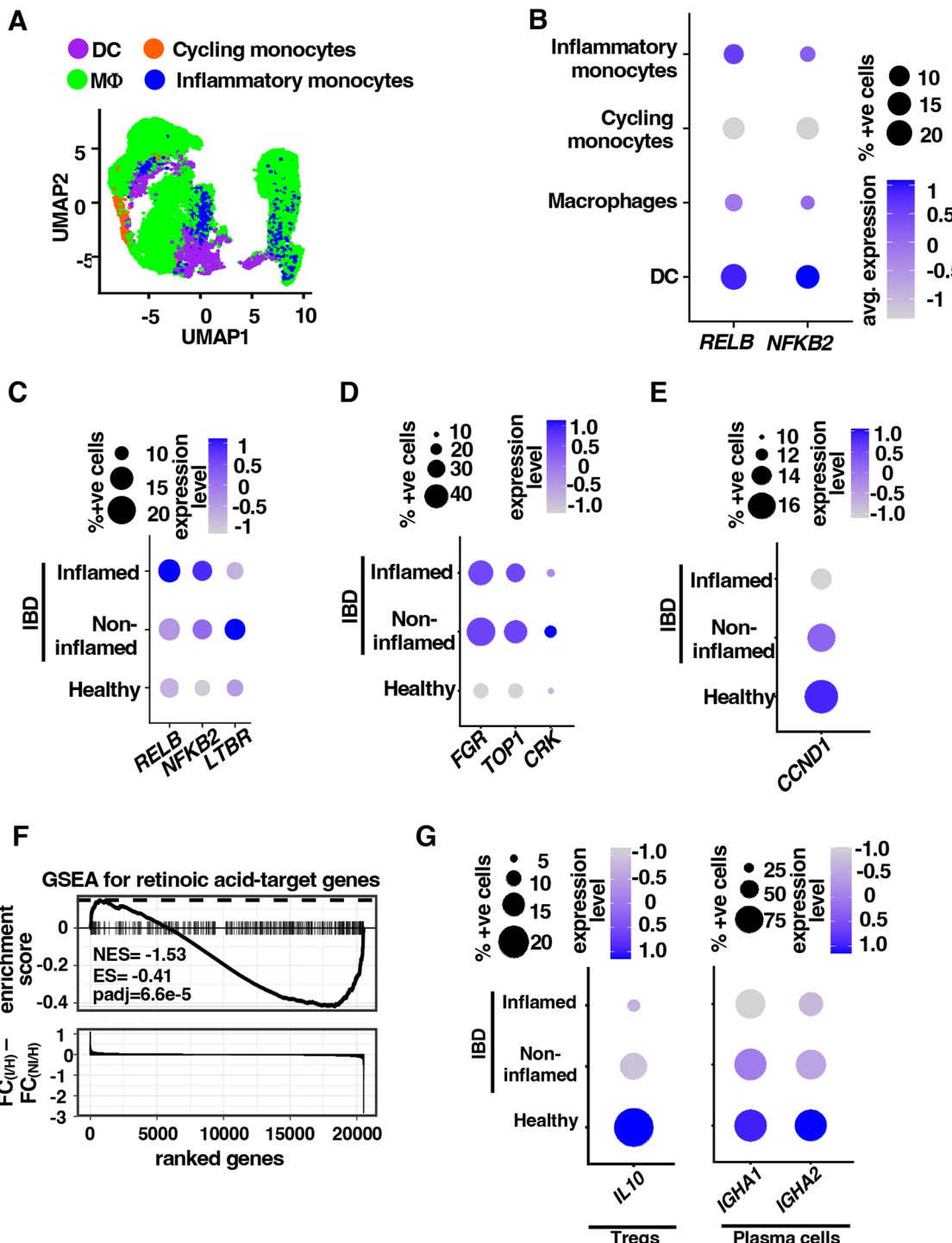

We further noticed interesting differences between *Relb* and *Nfkb2* genotypes. In particular, *Nfkb2*-deficient DCs did not show elevated IL-10 expressions as observed in RelB-deficient DCs. *Relb^{ΔCD11c}* mice accumulated intestinal IgA⁺ B cells, whereas *Nfkb2^{ΔCD11c}* mice did not. Unlike *Relb^{ΔCD11c}* mice, *Nfkb2^{ΔCD11c}* mice did not readily produce a phenotype in the DSS-induced acute colitis regime. Previous studies suggested that IL-10, in addition to DC-derived RA, promotes IgA class switching in the gut (Xu et al,

2007), and also documented an inhibitory role of RelB in DC-mediated IL-10 expressions (Duan et al, 2021). Additionally, RelB:p50 heterodimer was shown to partially compensate for the lack of RelB:p52 in the *Nfkb2*-deficient cell system (Basak et al, 2008). In light of these studies, we conjecture that RelB:p50 and RelB:p52 acted redundantly in DCs in downmodulating IL-10, while a non-redundant role of RelB:p52 in Axin1 transcription restrained β-catenin-dependent Raldh2 synthesis. Accordingly,

**Figure 7.   Engagement of noncanonical NF-κB-driven DC circuitry in human IBD.**

(A) Feature plot depicting DCs among indicated myeloid cells in colonic tissues from IBD patients and healthy individuals. Briefly, single-cell RNA-seq data available at Single Cell Portal was used for analyses (Data ref: Smillie et al, 2019). (B) Dot plot comparing the indicated cell types in the human gut for the expression of *RELB* and *NFKB2* mRNAs. (C) Dot plot showing the corresponding expression levels of the indicated genes in DCs as depicted in (C). The color gradient of dots represents the average expression level, while the dot size denotes the percentage of cells expressing a given gene. (D, E) Dot plot illustrating the expression of indicated transcripts in intestinal DCs. (F) GSEA comparing colonic biopsies from IBD patients to that of healthy subjects for the enrichment of RA targets among differentially expressed genes (top). Briefly, publicly available single-cell RNA-seq data (Data ref: Smillie et al, 2019) was analyzed for the expression of genes in inflamed and non-inflamed colonic segments from IBD patients. The difference in the fold changes (FC) in the mRNA levels between inflamed (I) and non-inflamed (NI) colonic segments with respect to the healthy subjects (H) were calculated for these genes, and the genes were ranked in descending order of the FC difference values (Bottom). Vertical lines represent individual RA targets. ES and NES denote enrichment and normalized enrichment scores, respectively. (G) Dot plot revealing the expression of *IL-10* in Tregs (left), and *IGHA1* and *IGHA2* in plasma cells (right) from the colon of IBD patients and healthy subjects.

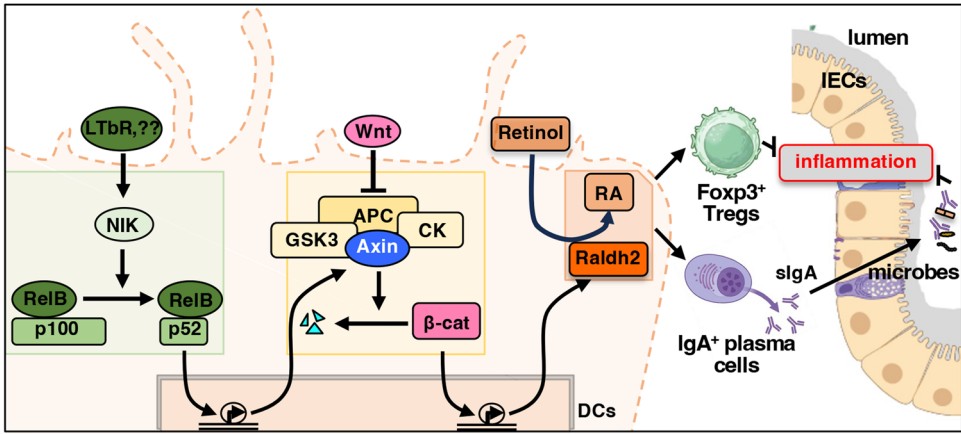

**Figure 8.   A cartoon depicting a noncanonical NF-κB pathway-driven signaling circuitry in DCs that tunes intestinal inflammation.**

ablating *Relb* in DCs supported the synergy between IL-10 and the RA pathway, culminating in the increased luminal IgA level. Conversely, RelB:p50 may have masked the phenotype of *Nfkb2*$^{\Delta CD11c}$ mice in the acute regime by preventing heightened IL-10 expressions in knockout DCs and, thereby, changes in the intestinal IgA⁺ cell compartment. *Nfkb2*$^{\Delta CD11c}$ mice also displayed a somewhat less marked increase in the frequency of intestinal Tregs compared to *Relb*$^{\Delta CD11c}$ mice, presumably owing to IL-10 expression differences between these genotypes. While the biochemical mechanism underlying RelB-mediated IL-10 control warrants further investigation, our comparison involving *Relb*$^{\Delta CD11c}$ and *Nfkb2*$^{\Delta CD11c}$ strains identified distinct as well as overlapping gene controls by the noncanonical NF-κB signal transducers *Relb* and *Nfkb2* in DCs with implications for intestinal health. Of note, no single animal model fully captures the clinical complexities of human IBD. Therefore, other models of experimental colitis, particularly the T-cell transfer model, should also be employed in the future to assess the generalizability of the proposed DC-intrinsic noncanonical NF-κB mechanisms in regulating intestinal inflammation.

The widely known role of the noncanonical NF-κB pathway lies in lymphoid stromal cells in supporting RelB-dependent expressions of homeostatic lymphokines, which promote secondary lymph node development during early embryogenesis and also lymphocyte ingress in lymph nodes in adults (Mukherjee et al, 2017a; Sun, 2017). In addition, *Nfkb2*-dependent crosstalk was

shown to modulate RelA-driven inflammatory gene expressions in a variety of cell types (Chawla et al, 2021; Shih et al, 2009; Basak et al, 2007). Although such cross-regulatory RelA controls by noncanonical NF-κB signaling are yet to be established in DCs, our work involving *Nfkb2*- as well as RelB-deficient cells unequivocally established that noncanonical RelB:p52 NF-κB signaling in DCs inhibited the tolerogenic β-catenin–Raldh2 axis, fueling intestinal pathologies. Further studies are required to determine if, independent of RelB, p100 directs immunogenic DC attributes via also RelA or another factor. Nevertheless, beyond tolerogenic DC functions, β-catenin and the Raldh2-RA pathway have other biological roles. For example, β-catenin deregulations in intestinal epithelial cells drive colon cancer (Zhao et al, 2022). In a recent study, Conlon et al suggested that LTβR inhibits WNT/β-catenin signaling in lung epithelial cells, exacerbating tissue damage in chronic obstructive pulmonary disease (Conlon et al, 2021). On the other hand, neuronal-derived RA promoted early lymph node biogenesis (Van De Pavert et al, 2009), and intestinal DC-secreted RA was critical for the maintenance of secondary lymph nodes (Zhang et al, 2016). We argue that our work may provide for a significant revision of our understanding of noncanonical NF-κB functions in physiology and disease. To this end, future studies ought to examine the potential trigger of noncanonical NF-κB signaling in various anatomic niches and their plausible impact on β-catenin activity or the RA pathway in shaping immune responses or developmental processes.

Finally, micronutrients are emerging as important determinants of mucosal immunity and potential interventional tools for inflammatory ailments (Gombart et al, 2020). Previous studies indicated that vitamin D exerts a negative impact on RelB synthesis in DCs (Ratra et al, 2022; Dong et al, 2005). As such, Raldh2 empowers intestinal DCs to secrete constitutively copious amounts of RA. Our current investigation posits that noncanonical NF-κB signaling curbs this immunomodulatory retinoic acid synthesis pathway in intestinal DCs, imparting vulnerability to colitogenic insults. Therefore, it appears that beyond its well-articulated role in immune gene expressions, DC-intrinsic noncanonical NF-κB signaling could be also critical for the micronutrient control of intestinal inflammation. While the significance of our study in the context of nutritional or DC-based therapy remains to be tested, we, in sum, establish a DC network integrating immune signaling and micronutrient metabolic pathways in the intestine.

# Methods

### Reagents and tools table

| Reagents | Source | Identifier/RRID |
| --- | --- | --- |
| **Antibodies** | | |
| Anti-RelA rabbit polyclonal | Santa Cruz Biotechnology | sc-372; AB_632037 |
| Anti-RelB rabbit polyclonal | Cell Signaling Technology | 4922S; AB_2179173 |
| Anti-mouse p52/p100 rabbit polyclonal | Generated in the Lab | Mouse p52/p100 sequence used: DNCY DPGLDGIPEYDD |
| Anti-Gapdh rabbit polyclonal | Cell Signaling Technology | 2118S; AB_10698756 |
| Anti-active β-catenin rabbit monoclonal | Cell Signaling Technology | 8814S; AB_11127203 |
| Anti-Raldh2 rabbit monoclonal | Cell Signaling Technology | 83805S; AB_2800032 |
| Anti-phospho-GSK-3β rabbit polyclonal | Cell Signaling Technology | 9336S; AB_331405 |
| Anti-GSK-3β rabbit polyclonal | Cell Signaling Technology | 9315S; AB_490890 |
| Anti-Axin1 rabbit monoclonal | Cell Signaling Technology | 2087S; AB_2274550 |
| Anti-Notch2 rabbit monoclonal | Cell Signaling Technology | 5732T; AB_10693319 |
| Anti-β-catenin Rabbit polyclonal | Invitrogen | 71-2700; AB_2533982 |
| Anti-IRF4 rabbit polyclonal | Thermo Scientific | PA5-21144; AB_11152871 |
| Goat anti-rabbit IgG-Cy5 secondary | Cytiva | PA45011; AB_772205 |
| Rabbit Trueblot Anti-Rabbit IgG HRP | Rockland | 18-8816-33; AB_2610848 |
| Anti-CD45.2 (104) BUV563 | eBiosciences | 365045482; AB_2925379 |
| Anti-CD25 (PC61.5) BUV661 | eBiosciences | 376025182; AB_2925446 |
| Anti-IL17A (17B7) BUV805 | eBiosciences | 368717382; AB_2896169 |
| Anti-CD197(CCR7) 4B12 eF450 | eBiosciences | 48197182; AB_1944351 |
| Anti-CD4(GK1.5) BUV737 | eBiosciences | 367004182; AB_2895921 |

| Reagents | Source | Identifier/RRID |
| --- | --- | --- |
| Anti-integrin a4b7 PE (DATK32) | eBiosciences | 12588782; AB_657803 |
| Anti-MHC II (M5/114.15.2) PE-Cy7 | eBiosciences | 25532182; AB_10870792 |
| Anti-CD64 (X54-5/7.1) PerCP-e710 | eBiosciences | 46064182; AB_2735016 |
| Anti-CD19 (1D3) BV780 | eBiosciences | 780109382; AB_2925722 |
| Anti-CD3E (145-2C11) BV780 | eBiosciences | 78003182; AB_2784894 |
| Anti-NK1.1(PK136) BV780 | eBiosciences | 78594182; AB_2744923 |
| Anti-Foxp3(150D/E4) FITC | eBiosciences | 11577382; AB_465243 |
| Anti-F4/80(BM8) FITC | eBiosciences | 123107; AB_103762287 |
| Anti-CD11b (M1/70) BV 510 | BioLegend | 101245; AB_2561390 |
| Anti-CD11c (N418) PerCP | BioLegend | 117236; AB_925727 |
| Anti-β-catenin1(15B8) PE | BioLegend | 862604; AB_2832863 |
| Anti-CCR9 (CW1.2) PE-Cy7 | BioLegend | 128712; AB_10901176 |
| Anti-IFNγ (XMG1.2) APC-Cy7 | BioLegend | 505850; AB_2616698 |
| Anti-B220 (RA3-6B2) FITC | BD Pharmingen | 553088; AB_394618 |
| Anti-IgA (mA-6E1) PE | eBiosciences | 12-4204-82; AB_465917 |
| Anti-CD103 (2E7) BUV737 | eBiosciences | 367-1031-82; AB_2895987 |
| **Chemicals and peptides** | | |
| DSS (36,000–50,000 MW) | MP Biomedicals | 160110 |
| FITC-Dextran (MW 3000 - 5000) | Sigma-Aldrich | 60842-46-8 |
| Collagenase, Type 4 | Gibco | 17104019 |
| DNase I | Worthington | LS002139 |
| LPS from *E. coli* serotype O55:B5 | Enzo Life Sciences | ALX-581-013-L002 |
| Porcn Inhibitor-II, C59-inhibitor | Merck | SKU5004960001 |
| Wnt-β-catenin signaling inhibitor -Fzm1 | Merck | 534358 |
| BMS493 (RAR inhibitor) | MedChem Express | HY-108529 |
| β-catenin/Tcf Inhibitor III, iCRT3 | Sigma-Aldrich | 219332 |
| LTβR-IgG fusion protein | Biogen Inc. | US20110046073A1 |
| Recombinant mouse TGFβ1 | BioLegend | 763104 |
| Anti-CD3 | Thermo Scientific | 16-0032-86; AB_467057 |
| Anti-CD28 | Thermo Scientific | 16-0281-86; AB_468923 |
| Recombinant mouse GM-CSF | Miltenyi Biotec | 130095739 |
| Recombinant mouse IL-4 | Miltenyi Biotec | 130097757 |
| **Commercial kits** | | |
| RNeasy Mini Kit (250) | Qiagen | 74106 |
| Primescript 1st strand cDNA kit | Takara Bio | 6110B |
| Foxp3/TF Staining Buffer | Invitrogen | 00-5523-00 |
| ALDEFLUOR Kit | Stem cell technologies | 01700 |
| Leukocyte activation kit | BD Biosciences | 550583 |
| Naive CD4 T-cell isolation kit | Miltenyi Biotec | 130 104 453 |
| Live/dead fixation yellow dye | Invitrogen | L34968 |
| QIAamp PowerFecal Pro DNA kit | Qiagen | 51840 |
| Mouse IgA Uncoated ELISA kit | Invitrogen | 88-50450-22 |

| Reagents | Source | Identifier/RRID |
|---|---|---|
| **Mice models** | | |
| *Nfkb2*$^{-/-}$ | small animal facility, NII | N/A |
| *Relb*$^{fl/fl}$ mice | Jackson Laboratories | 028719 |
| *Nfkb2*$^{fl/fl}$ mice | Jackson Laboratories | 028720 |
| *Ctnnb1* $^{fl/fl}$ mice (floxed for β-catenin gene) | generous gift from Prof. Amitabha Bandopadhyay, IIT Kanpur | N/A |
| *Itgax*-Cre mice (CD11c-Cre) | Jackson Laboratories | 008068 |
| **Primers for qRT-PCR (mouse genes)** | | |
| **Gene** | **Forward (5′–3′)** | **Reverse (5′–3′)** |
| *Il1b* | AACCTGCTGGTGTGTGACGTTC | CAGCACGAGGCTTTTTTGTTGT |
| *Cxcl10* | ACCAACCACCAGGCTAGA | GCGTCACACTCAAGCTCT |
| *Aldh1a2* | ATGGGTGAGTTTGGCTTACG | GGTTCATTGGAAGGCAGAAA |
| *Il10a* | CTAACCGACTCCTTAATGC | AATCACTCTTCACCTGCTC |
| *Actin* | CCAACCGTGAAAAGATGAC | GTACGACCAGAGGCATACAG |
| *Ctnnb1* | GTTCGCCTTCATTATGGACTGCC | ATAGCACCCTGTTCCCGCAAAG |
| *Axin1* | CACCCAGAAGCTGCTATTGGAGA | CCAGGGCATAGCCAGAGTTGA |
| **Bacterial taxa primers** | | |
| *Actinobacteria* | CGCGGCCTATCAGCTTGTTG | ATTACCGCGGCTGCTGG |
| *Firmicutes* | GGAGYATGTGGTTTAATTCGAAGCA | AGCTGACGACAACCATGCAC |
| *Bifidobacterium* | TCGCGTCCGGTGTGAAAG | CCACATCCAGCATCCAC |
| *Enterobacter* | GTGCCAGCMGCCGCGGTAA | GCCTCAAGGGCACAACCTCCAAG |
| *SFB* | GACGCTGAGGCATGAGAGCA | GACGGCACGGATTGTTATTC |
| *Sutterella* | CGCGAAAAACCTTACCTAGCC | GACGTGTGAGGCCCTAGCC |
| *Prevotella* | CACGGTAAACGATGGATGCC | GGTCGGGTTGCAGACC |
| *β-Proteobacteria* | TCACTGCTACACGYG | ACTCCTACGGGAGGCAGCAG |
| *Universal 16s* | TCCTACGGGAGGCAGCAGT | GGACTACCAGGGTATCTAATCCTGTT |
| **Software/computational tools** | | |
| **Tools** | **Source** | |
| DESeq2 package | https://bioconductor.org/packages/release/bioc/html/DESeq2.html (Love et al, 2014) | |
| fgsea package | https://bioconductor.org/packages/release/bioc/html/fgsea.html (Korotkevich et al) | |
| Integrative Genomics Viewer v2.16.2 | https://igv.org/app/ (Robinson et al, 2011) | |
| MEME Suite | https://meme-suite.org/meme/tools/meme (Bailey et al, 2015) | |

| Reagents | Source | Identifier/RRID |
|---|---|---|
| AUCell package | https://bioconductor.org/packages/release/bioc/html/AUCell.html (Aibar et al, 2017) | |
| Seurat package | https://satijalab.org/seurat/reference/seurat-package (Satija et al, 2015) | |
| fastQC | https://www.bioinformatics.babraham.ac.uk/projects/fastqc/ | |
| Paired-End reAd merger (PEAR) | https://www.h-its.org/downloads/pear-academic/ (Zhang et al, 2014) | |
| QIIME2-2022.2 | https://qiime2.org/ (Guerrini et al, 2019) | |
| SILVA SSU 138 | https://www.arb-silva.de/documentation/release-1381/ (Pruesse et al, 2007) | |
| ampvis R-package | https://kasperskytte.github.io/ampvis2/articles/ampvis2.html | |
| ggplot2 package | https://ggplot2.tidyverse.org/ (Wickham, 2011) | |
| Biorender | https://www.biorender.com/ | |

## Methods and protocols

### Animal use

Sources of parental mouse strains used in this study have been indicated in the Reagents and Tools Table. All strains were housed at the National Institute of Immunology in the specific pathogen-free facility, and used in accordance with the guidelines of the institutional animal ethics committee (Approval no.—IAEC – 487/18, 615/22, IBSC-512/22). CD11c-Cre mice were crossed with various floxed genotypes for generating CD11c cell-specific knockouts. Mice were allocated randomly to experimental groups and no blinding was done.

### Studying chemically induced colitis in mice

Seven to eight weeks old littermate male mice of indicated genotypes were cohoused for at least 1 week prior to experiments. As described (Wirtz et al, 2017), acute colitis was induced by administering 1.5% DSS in drinking water for 7 days, otherwise indicated. For some experiments, mice were peritoneally injected with 200 µg of LTβR-IgG fusion protein on −1, 2, and 4 days of DSS-administration. For chronic colitis induction, mice were fed with 1% DSS in drinking water for 7 days followed by 13 days of normal water; this cycle was repeated thrice. DSS-treated mice were examined for bodyweight changes in a time course. As a humane endpoint, mice with more than 20% bodyweight loss were immediately euthanized. For certain sets, mice were euthanized

on indicated days post-DSS treatment, and the length of the excised colon was measured. For histological studies, the colon was fixed in 4% formalin and embedded onto the paraffin block following the Swiss roll technique. Colon sections were examined following hematoxylin and eosin (H&E) staining or upon additional staining using Alcian Blue. Images were captured in an Olympus inverted microscope using Image-Pro6 software. To assess intestinal permeability, mice were gavaged with 440 mg/kg FITC-dextran 6 h before fluorescence measurements in sera.

## Analyses of colonic immune cells

LP immune cells were isolated from dissected mouse colons, as described (Kim et al, 2022), except that an 80 and 40% Percoll gradient was performed and lymphocytes were collected from the interphase of Percoll layers. For MLNs and the spleen, single-cell suspensions were prepared by meshing these organs using a 70-micron cell strainer. Cells were then stained with fluorochrome-conjugated antibodies, and analyzed by flow cytometry. For surface staining, cells were labeled with antibodies in the staining buffer for 30 min on ice. Intracellular staining was performed using a commercially available kit as per the manufacturer's instruction (eBiosciences). For T-cell analysis, cells were stimulated using leukocyte activation cocktail (BD Biosciences) for 4 h before being subjected to staining. Flow cytometry data were acquired on A5 symphony and analyzed using FlowJo v10 software. For sorting cells, BD FACSAria III cell sorter was used. Details of the gating strategy and flow cytometry antibodies have been provided in Appendix Fig. S3A and Reagents and Tools Table, respectively.

## Studies involving BMDCs

BMDCs were generated from mouse bone marrow cells following a revised GM-CSF/IL-4 protocol that involves supplementing GM-CSF-containing DC-differentiation medium with rIL-4 but from day 6 of cell seeding (Jin and Sprent, 2018). Non-adherent cells were collected from the culture at day nine and seeded in fresh plates for experiments. BMDC differentiation was confirmed by examining the surface expression of CD11c and MHC II by flow cytometry. Bone marrow cells from $Relb^{\Delta CD11c}$ and $Nfkb2^{\Delta CD11c}$ mice were used to produce BMDCs lacking the function of $Relb$ and $Nfkb2$, respectively. Gene disruptions were confirmed in these day-9 BMDCs by immunoblot analyses.

For the Treg generation assay, naive CD4$^+$ T cells (CD4$^+$CD25$^-$CD62L$^+$) were sorted from the spleen of WT C57Bl/6 mice using a commercially available kit (Miltenyi Biotec). Next, $1 \times 10^5$ naive CD4$^+$ T cells were co-cultured with WT or gene-deficient BMDCs at a ratio of 1:1 in the presence of a soluble anti-CD3 antibody (1 µg/ml) and Treg polarizing cytokines—TGF-β (5 ng/ml), rIL2 (20 U/ml), and RAR inhibitor, BMS493 (100 nM) in a round-bottom 96-well plate. After another 3.5 days in the culture, cells were subjected to flow cytometry analyses for the expression of CD4, CD25, and FoxP3. RALDH activity was determined using the ALDEFLUOR assay kit, as per the manufacturer's protocol (Stem Cell Technologies). Specifically, 0.5–1 million cells per ml were used for the analysis and incubated for 35 mins. The frequency of Aldefluor$^+$ cells in the CD11c$^+$MHCII$^{int/hi}$ BMDC population was scored by flow cytometry.

## Biochemical and gene expression studies

Immunoblot analyses and RT-qPCR were performed essentially following our previously published methods (Chawla et al, 2021) using WT or gene-deficient BMDCs. In certain experiments, BMDCs were also treated with 100 ng/ml of LPS for 8 h. For in vivo analyses, cells were sorted from mice MLN using antibodies against required markers and similarly processed for immunoblotting. Primary antibodies and RT-qPCR primers used in this study have been described in the Reagents and Tools Table. For Immunoblot assays, Cy5 conjugated secondary antibodies were used (Cytiva Lifesciences); gel images were acquired using Typhoon Variable Mode Imager, and band intensities were quantified in ImageQuant. For detecting active β-catenin, an antibody (CST, 8814 S) that specifically recognizes unphosphorylated β-catenin simultaneously at Ser-33, Ser-37, and Thr-41 was used. As described earlier (Mukherjee et al, 2017b), whole-cell lysate prepared from ~10$^6$ cells was used for co-immunoprecipitation studies. Immunoprecipitation was performed using 2 µg of anti-β-catenin antibody, and immunopellets were examined by immuno-blot analyses using TrueBlot secondary antibody. For colonic mRNA measurements, total RNA isolated from tissues derived from the distal colon was used.

For RNA-seq studies, BMDCs generated from WT and global $Nfkb2^{-/-}$ mice were compared in triplicates. Library preparation and sequencing were carried out at Nucleome Informatics, Hyderabad. When the cumulative read count of a transcript estimated from a total of six experimental sets was less than fifty, the corresponding gene was excluded from the analysis. Ensembl IDs lacking assigned gene names were also excluded. Accordingly, we arrived at a list of 19118 genes. Next, average read counts of these genes in experimental sets were calculated, and fold changes in the mRNA levels in $Nfkb2^{-/-}$ cells in relation to WT BMDCs were determined using the DEseq2 package from Bioconductor. Genes were then ranked in descending order of this fold change difference. Differentially expressed genes with fold change values above or below 1.2 were subjected to the pathway enrichment analysis using WikiPathways subset of cellular processes available at MSigDB (www.gseamsigdb.org). The ranked gene list was also examined by GSEA involving the fgsea package using a previously described list of Retinoic acid targets (Balmer and Blomhoff, 2002). Integrative Genome viewer was used for generating genome browser tracks for RelB binding to the $Axin1$ locus. For identifying NF-κB binding motifs around the ChIP-seq peaks, we used the Meme Suite.

## Single-cell RNA-Seq data analysis

We analyzed publicly available single-cell RNA-Seq data generated using colonoscopic biopsies from human IBD patients (Smillie et al, 2019; Devlin et al, 2021; Data ref: Smillie et al, 2019; Devlin et al, 2021), and those derived using colonic tissues from DSS-treated WT mice (Ho et al, 2021; Data ref: Ho et al, 2021) For human IBD, a total of 1819 colonic DCs from healthy or inflamed biopsies were examined. For DSS-induced colitis in mice, a total of 2236 intestinal mononuclear phagocytes (MNPs) from either untreated WT mice or those subjected to DSS treatment for 3 or 6 days were collectively considered. Leveraging a list of prototypic macrophage or DC genes, these MNPs were clustered into macrophages or DCs using the AUCell package in R. Then intestinal DCs were examined for the expression of indicated genes. Seurat package in R was used for gene expression analyses.

## Microbiome studies

Fresh feces were collected from 8-week-old littermate floxed and gene-deficient mice that were cohoused. Bacterial genomic DNA was extracted using QIAamp PowerFecal Pro DNA kit (Qiagen). The V3-V4 region of 16s rDNA was amplified by PCR and sequenced in the MiSeq platform at THSTI (Illumina) using the $2 \times 250$ bp paired-end protocol yielding pair-end reads that overlap almost completely. The read quality was examined by fastQC program (see Reagents and Tools Table for program source). Reads were processed based on a threshold phred score (Q phred) ≥20, and then merged to obtain high quality, long single reads using Paired-End reAd mergeR (PEAR) program (see Reagents and Tools Table for program source). The QIIME2 pipeline (qiime2-2022.2) was used for generating Operational Taxonomic Unit (OTU) table from the merged reads—reads were clustered based on 99% sequence identity. The naive Bayes machine-learning classifier plugin of QIIME2's q2-feature-classifier was used to find taxonomic classification up to the genus level based on SILVA SSU 138 (See Reagents and Tools Table for program source). The core OTU was subtracted from the main OTU table if an OTU was present in ≤80% of the total sample of the group. The relative abundance of each OTU was calculated by number of the OTU in each sample divided by total number of OTUs in all the samples. The differential microbial abundance was converted into log 10 values to show negatively and positively abundant taxa in both groups. The microbial alpha diversity analysis was performed using the ampvis2 R-package program. The statistical analysis was carried out using the tidyverse and ggplot2 R-packages (see Reagents and Tools Table for program source).

Further, the composition of bacterial taxa in flow cytometry-sorted IgA+ and IgA− fractions of the fecal microbiota was assessed using qPCR with the primers described in Reagents and Tools Table. For measuring fecal IgA by ELISA, feces were homogenized in sterile PBS, and centrifuged to remove bacteria and insoluble debris. Fecal samples were then serially diluted and analyzed for IgA concentration by ELISA using a commercially available kit following the manufacturer's protocol (Invitrogen). Reading was acquired in a microplate reader (MultiSkan FC microplate photometer).

## Statistical analysis

All experiments undergoing statistical tests were performed at least thrice or more as biological replicates. No data were excluded from the analyses. GraphPad Prism 10 was used to plot and perform the statistical analysis. Statistical analyses were performed using unpaired $t$ test for comparison of two groups and one-way ANOVA followed by Tukey's multiple comparisons test for multiple groups indicated in the respective figure legends. All quantitative data are denoted as mean ± standard error of mean (SEM). No statistical methods were used to the sample size.

## Data availability

The MIAME version of the RNA-seq dataset is available on NCBI Gene Expression Omnibus accession GSE249101, and the microbiome data is available at the accession PRJNA1046443. All the raw data is available at BioStudies accession ID: S-BSST1441.

The source data of this paper are collected in the following database record: biostudies:S-SCDT-10_1038-S44318-024-00182-6.

## Peer review information

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

## Acknowledgements

The authors thank Prof. Amitabha Bandyopadhyay, IIT Kanpur for help with *Ctnnb* $^{fl/fl}$ mice. The authors thank Biogen Inc. for providing us with LTβR-IgG fusion protein. The authors thank V. Kumar for technical support and Dr. P. Nagarajan for the help with animal husbandry. Research in the PI's laboratory was funded by DBT (BT/PR36631/BRB/10/ 1862/2020) and NII-Core. AD and NK thank DST-INSPIRE and DBT, respectively, for research fellowships.

## Author contributions

**Alvina Deka**: Conceptualization; Data curation; Formal analysis; Investigation; Visualization; Methodology; Writing—original draft; Writing—review and editing. **Naveen Kumar**: Conceptualization; Data curation; Formal analysis; Investigation; Visualization; Methodology; Writing—original draft; Writing—review and editing; Large scale data analysis. **Swapnava Basu**: Investigation; Large scale data analysis. **Meenakshi Chawla**: Investigation; Methodology. **Namrata Bhattacharya**: Methodology; Large scale data analysis. **Sk Asif Ali**: Investigation. **Bhawna**: Investigation. **Upasna Madan**: Investigation; Methodology. **Shakti Kumar**: Methodology; Data analysis of microbiome studies. **Bhabatosh Das**: Supervision. **Debarka Sengupta**: Supervision. **Amit Awasthi**: Supervision. **Soumen Basak**: Conceptualization; Formal analysis; Supervision; Funding acquisition; Validation; Writing—original draft; Project administration; Writing—review and editing.

Source data underlying figure panels in this paper may have individual authorship assigned. Where available, figure panel/source data authorship is listed in the following database record: biostudies:S-SCDT-10_1038-S44318-024-00182-6.

## Disclosure and competing interests statement

The authors declare no competing interests.

