## [Peer Review File · The EMBO Journal]

Non-canonical NF- κ B signaling limits the tolerogenic β -catenin-Raldh2 axis in gut dendritic cells to exacerbate intestinal pathologies

Soumen Basak, Alvina Deka, Naveen Kumar, Swapnava Basu, Meenakshi Chawla, Namrata Bhattacharya, Sk Ali, Bhawna ., Upasna Madan, Shakti Kumar, Bhabatosh Das, Debarka Sengupta, and Amit Awasthi

Corresponding author(s): Soumen Basak (sobasak@nii.ac.in)

Review Timeline:

Transfer from Review Commons:	31st Mar 24
Editorial Decision:	15th Apr 24
Revision Received:	4th Jun 24
Editorial Decision:	5th Jul 24
Revision Received:	9th Jul 24
Editorial Decision:	11th Jul 24
Revision Received:	12th Jul 24
Accepted:	12th Jul 24

Editor: Ioannis Papaioannou

Transaction Report:

Review #1

1. Evidence, reproducibility and clarity:

Evidence, reproducibility and clarity (Required)

Deka and colleagues report that a non-canonical NF κ B signaling operates in DCs in the context of inflammation and inhibits a tolerogenic mechanism driven by β -catenin-Raldh2. The following comments are made to clarify the findings presented.

1. The authors re-analyzed published scRNAseq data from DSS colitis to identify the expression of Relb and NF κ B2 in myeloid cells.
 - a. The authors are encouraged to expand this analysis to other published datasets.
 - b. Additionally, the expression of Relb and NF κ B2 in other cells - especially other myeloid cells - should be explored and included, even if then the authors later choose to test their function in DCs.
 - c. Please note there is a transition from Fig 1A to Fig1B to focus on DCs, which is not apparent from the figure.
2. Please include scale bars for all histological analyses.
3. In Fig S1I, the authors show that loss of body weight upon DSS treatment in Nfkb2deltaCD11c is indistinguishable from control. Why is the starting weight at 110%? Please clarify.
4. In Figure 2, please indicate the database/s used for identification of top biological pathways.
5. The authors show a more significant expansion in Tregs upon DSS treatment when non-canonical NF κ B is ablated in DCs. Is this at the expense of a reduction of specific Th cells? Can the authors also report the number of cells, beyond the % of cells?
6. In figure 6A, it appears that not only the amount of beta-catenin expressed, but also the percentage of beta-catenin positive MNL DCs is significantly expanded upon ablation of non-canonical NF κ B. Please verify and if so, include.
7. In analogy to the comment #1 above, please expand the analyses in human samples to include the expression of Relb and Nfkb2 to other myeloid cells.

2. Significance:

Significance (Required)

Strengths of the manuscript include the conceptual novelty of the intersection between non-canonical NF κ B and the tolerogenic β -catenin-Raldh2 axis. And additional strength is the methodic approach, which includes various immunological and

biochemical assessments as well as genetic perturbations to dissect such relationships. While it remains unknown the relevant triggers for the non-canonical axis described, this study advances our mechanistic understanding on how activation of this axis overrides regulatory mechanisms in DCs. As such, this manuscript should be of broad interest to immunologists and in particular mucosal immunologists.

3. How much time do you estimate the authors will need to complete the suggested revisions:

Estimated time to Complete Revisions (Required)

(Decision Recommendation)

Between 1 and 3 months

No

Review #2

1. Evidence, reproducibility and clarity:

Evidence, reproducibility and clarity (Required)

The following issues are noted.

- all animal strains and their provenance should be described and properly referenced (for example, there are at least two CD11c-Cre strains with different specificity). Along the same lines, the specificity of Cre recombination should be confirmed, at least in major cell types (DCs vs T effector or regulatory cells).

- the DSS model is prone to "batch effects" of individual cages, and proper comparison between genotypes is possible only if mice of different genotypes (eg littermates) are housed together in the same cages. The authors should clearly confirm whether this was the case, and if not, key experiments should be repeated in this setting.
- BMDCs represent a heterogeneous mixture of DCs and macrophages (Helft et al., Immunity 2015). These populations should be clearly defined and compared between genotypes, to make sure that they do not underlie the observed gene expression differences.
- the analysis of DCs in mutant strains (e.g. in Fig. 3) would benefit from better definition of populations, e.g. resident vs migratory DCs in the MLN, the Notch2-dependent CD103+ CD11b+ DCs in the LP and MLN, etc. Again, this would be important to justify differences in gene expression (e.g. Fig. 3D).
- the analysis of b-catenin protein expression and cellular localization at single-cell level (e.g. by IF) would greatly strengthen the mechanistic connection between NF-kB and Wnt/b-catenin pathways.

****Minor:****

- the reanalyses of previous single-cell data in Figs. 1 and 7 are much less convincing or exciting than the new experimental data relegated to the supplements. The distribution of the results between main and experimental figures may be reconsidered in this light.

2. Significance:

Significance (Required)

The manuscript by Deka et al. explores the role of the non-canonical NF-kB pathway, specifically of its key mediators RelB and NF-kB2, in dendritic cells (DCs) during intestinal inflammation. The key strength of the paper is the demonstration that DC-specific deletion of RelB or NF-kB2 lead to improved acute or chronic DSS colitis. It is also shown that reducing the dose of b-catenin rescues the phenotype of RelB deletion, providing an important genetic connection between NF-kB and Wnt/b-catenin pathways. As such, the work is novel, important and of potential significance to the field.

3. How much time do you estimate the authors will need to complete the suggested revisions:

Estimated time to Complete Revisions (Required)

(Decision Recommendation)

Between 1 and 3 months

No

Review #3

1. Evidence, reproducibility and clarity:

Evidence, reproducibility and clarity (Required)

****Summary:****

This manuscript from Deka et al. investigates the role of dendritic cell noncanonical NF κ B signaling on intestinal inflammation. Based on prior data showing altered DC function in intestinal inflammation, they interrogated existing scRNAseq data and found that DSS treatment (which yields a chemical colitis) increased expression of non-canonical NF κ B family members in dendritic cells. This led to the generation of a DC specific RelB deficient mouse and use of a DC specific NF κ B2 deficient mouse, each of which showed varying degrees of protection from chemical colitis. Overall, they do a very nice job identifying a mechanism by which noncanonical NF κ B signaling in dendritic cells contributes to intestinal inflammation via transcriptional regulation of Axin1, downregulation of β -catenin, restraint of Raldh2 synthesis, impaired retinoic acid synthesis and subsequent decrease in protective Tregs, IgA+ B cells, and microbial dysbiosis. The importance of this pathway is well supported by their focused targeting of β -catenin. After pharmacologic inhibition of β -catenin showed restoration of Raldh2 abundance, they made a DC specific β -catenin haploinsufficiency RelB Δ CD11c mouse which showed impaired Raldh2 activity with restoration of colonic Tregs and fecal sIgA. When challenged with DSS, the

protective phenotype seen with the RelB Δ CD11c was lost and the colitis phenotype returned to that of the Relbfl/fl control, further solidifying the role of β -catenin, Raldh2 and RA on intestinal inflammation.

Additionally, the discussion provides a robust mechanistic explanation for the phenotypic differences between the RelB and Nfkb2 genotypes, drawing on the authors' deep knowledge of the non-canonical NF κ B pathway.

****Major comments:****

1. Although they propose a novel mechanism by which dendritic cells can contribute to intestinal inflammation, it is in a model of acute epithelial injury that accentuates the contribution of the innate immune system. Would recommend including a discussion of the limitations of this model.
2. The human work (Figure 7) shows solid evidence of heightened non-canonical NF κ B signaling in DCs via abundance of RELB and NF κ B2 along with a few RelB important genes, however the RA specific pathway identified in the mouse work is not strongly corroborated by the human data. There is demonstration of one β -activated gene (CCDN1) showing decreased expression in IBD patients, however no other gene along with RA pathway was clearly identified to be differentially expressed as one would predict from the mouse work.

****Minor comments:****

1. Their NF κ B2 Δ CD11c mouse underwent a regimen of chronic DSS treatment after acute DSS treatment only displayed subtle phenotypic changes. Was the same chronic colitis regimen also tested in the RelB Δ CD11c ?
2. In the introduction, it was stated that patients with UC have a marked reduction in intestinal DCs. If DCs (particularly non-canonical NF κ B signaling) promote inflammation, how do you explain a decrease in this cell type in patients with active disease?
3. The focus on retinoic acid is interesting, however may be oversimplifying the role of non-canonical NF κ B in DCs on the mucosal immune system. It must also be mentioned that there is crosstalk between the non-canonical and canonical NF κ B signaling systems, for example Nfkb2 is capable of functioning as a I κ B protein and inhibiting RelA-p50 (from the last author's prior work - Basak et al, Cell, 2007). Thus would include some mention of possible effects on the canonical system that contribute to intestinal inflammation.
4. In the single cell DSS data they analyzed, there was a distinct DC population was seen with DSS colitis treatment. Although they are categorized as cDC2s, what genes separate them from the other DC populations?
5. Why was the RNAseq work on BMDCs (that identified RA metabolism as a top

ranking differentially expressed pathway) done only on Nfkb-/- BMDCs and not RelB-/-? The RelB-/- had a more pronounced protected phenotype in the cell type specific knockout, and is a cleaner target (does not have the I κ B capability of Nfkb2).

2. Significance:

Significance (Required)

As a physician scientist who clinically cares for patients with inflammatory bowel disease, and scientifically studies signaling within innate immune cells, this manuscript does a rigorous job of identifying a mechanism by which canonical NF κ B signaling in dendritic cells contributes to intestinal inflammation. This study would be very informative for both basic and translational researchers as it identifies a clear pathway by which the innate immune system contributes to intestinal inflammation, and opens up room for inquiry into triggers of non-canonical NF κ B in IBD and modulation of the RA pathway as a potential novel therapeutic target.

3. How much time do you estimate the authors will need to complete the suggested revisions:

Estimated time to Complete Revisions (Required)

(Decision Recommendation)

Less than 1 month

Yes

Revision Plan

Manuscript number: RC-2024-02368R

Corresponding author(s): Soumen Basak

1. General Statements [optional]

We sincerely thank the Editors and the knowledgeable reviewers for the valuable suggestions, constructive critiques, and enthusiasm about the work described in our initial submission entitled "Non-canonical NF- κ B signaling promotes gut inflammation by restraining the tolerogenic β -catenin-Raldh2 axis in dendritic cells".

As it is perceived, all three referees were very supportive of our study. In fact, they lauded our work as conceptually novel, methodical, rigorous and of potential significance to the field. They submitted that our findings may potentially inform novel therapeutic targets in inflammatory ailments in the future. However, the reviewers also offered several important suggestions for further validating the proposed mechanistic model and improving the scientific content of the draft. In a meticulously revised manuscript, we have now presented additional data in the results section and also further elaborated signalling crosstalk mechanisms in the discussion section (Please see point-by-point responses to reviewers' comments and revised draft). Our revised version indeed aptly upholds our conclusion that non-canonical NF- κ B signaling in DCs fuels aberrant gut inflammation by modulating the RA pathway. We admit that this rigorous revision has profoundly improved the overall quality of the science presented in our manuscript.

2. Description of the planned revisions

Reviewer #2

The analysis of DCs in mutant strains (e.g. in Fig. 3) would benefit from a better definition of populations, e.g. resident vs migratory DCs in the MLN, the Notch2-dependent CD103+ CD11b+ DCs in the LP and MLN, etc. Again, this would be important to justify differences in gene expression (e.g. Fig. 3D).

We sincerely appreciate the comment from the knowledgeable reviewer. In a landmark paper from Prof. Fiona Powrie's group (Coombes et al., 2007), it was earlier demonstrated that CD103+ DCs present in the intestine migrate to local MLNs and play a key role in producing RA and supporting Tregs. While our BMDC data strongly supported a cell-intrinsic mechanism underlying Raldh2 upregulation upon non-canonical NF- κ B deficiency (Figure 2F), our *in vivo* studies (Figure 3D-3E) did not entirely rule out also a possible expansion of RA-producing CD103+ DC compartment in our knockout mice. Although the proposition that non-canonical NF- κ B signaling regulates the generation of specific intestinal DC subsets seems attractive, we must point out that previous studies showed a relatively unaltered frequency of CD103+ cells among steady-state migratory DCs in skin-draining lymph nodes (Döhler et al., 2017). Nevertheless, following the reviewer's suggestion, we now plan to perform advanced flow cytometry analyses to compare *Relb*^{fl/fl} and *Relb*^{ACD11c} mice for the frequency of CD103+CD11b-, CD103+CD11b+ and CD103-CD11b+ DCs in the intestine. To this end, we have already optimized our experimental protocol for staining intestinal DCs with anti-CD103 antibody (BD Bioscience). In the coming weeks, we are expecting to gather adequate numbers of littermate knockout mice to perform a side-by-side comparison.

3. Description of the revisions that have already been incorporated in the transferred manuscript

Reviewer #1 (Evidence, reproducibility and clarity (Required)):

Deka and colleagues report that a non-canonical NFKb signaling operates in DCs in the context of inflammation and inhibits a tolerogenic mechanism driven by b-catenin-Raldh2. The following comments are made to clarify the findings presented.

1. The authors re-analyzed published scRNAseq data from DSS colitis to identify the expression of *Relb* and *NFKB2* in myeloid cells.

a. The authors are encouraged to expand this analysis to other published datasets.

We sincerely appreciate the comment from the knowledgeable reviewer. Unfortunately, we did not find any other publicly available scRNAseq dataset from DSS-treated mice. To circumvent this problem, we instead examined a previously published microarray-based bulk transcriptomic dataset obtained using FACS-sorted DCs isolated from the mouse colon (GSE58446, Muzaki et al. 2016; doi:10.1038/mi.2015.64). We consistently found an increased expression of *Relb*, and to some extent *Nfkb2*, mRNAs in intestinal DCs upon DSS treatment (Fig R1). Because microarray analysis lacks the quantitative attributes that scRNAseq offers, we provided this newly analysed dataset for reviewer's eyes and refrained from including this data in the manuscript *per se*. Of note, we have also provided our own experimental data directly demonstrating p100 processing in DC isolated from colitogenic gut. (Fig 1F)

Figure R1: Heatmap revealing the expression of *Relb* and *Nfkb2* mRNA in gut DCs. Briefly, publicly available microarray dataset (Ref: GSE58446) was used; DCs from the colonic lamina propria of wild-type BALB/c, untreated or administered with 2% DSS for 4 days, were analyzed for gene expression.

Importantly, we could identify an additional scRNAseq dataset derived colitogenic human ulcerative colitis patients (GSE162335, Devlin et al. 2021, doi.org/10.1053/j.gastro.2020.12.030). Interrogation of this dataset indeed confirmed that *RELB* and *NFKB2* mRNAs were majorly expressed in intestinal DCs and not in intestinal macrophages and that IBD was associated with increased expression of multiple *RelB*-important genes in intestinal DCs. These analyses further supported the notion that heightened non-canonical NF- κ B signalling in DCs could be fuelling aberrant gut inflammation. We have now incorporated this newly acquired data in the supplementary Figure S5A-S5C. (line# 456-460)

b. Additionally, the expression of *Relb* and *NFKB2* in other cells - especially other myeloid cells - should be explored and included, even if then the authors later choose to test their function in DCs.

Adhering to this brilliant suggestion, we have now further interrogated the mouse scRNAseq dataset (GSE148794 ; Ho et al., 2021) to compare macrophages and DCs for the

Revision Plan

expression of the non-canonical signal transducers. Indeed, we found a relatively insignificant level of *Relb* and *Nfkb2* mRNAs in intestinal macrophages in comparison to intestinal DCs. Our data suggested that the non-canonical NF- κ B pathway is likely to play a more prominent role in DCs than in macrophages in the gut. This new analysis has now been presented in the revised draft in Figure 1B. (revised text line#136-140) This comparison indeed proved useful in motivating subsequent in-depth analyses of the non-canonical NF- κ B pathway in DCs in the context of experimental colitis.

c. Please note that there is a transition from Fig 1A to Fig 1B to focus on DCs, which is not apparent from the figure.

Please find our response to #1b.

2. Please include scale bars for all histological analyses.

We thank the reviewer for alerting us. The scale bars were already included in the histological analyses; we have now appropriately highlighted them in this revised version for better visual clarity.

3. In Fig S1I, the authors show that loss of body weight upon DSS treatment in *Nfkb2*^{ADC11c} is indistinguishable from control. Why is the starting weight at 110%? Please clarify.

We sincerely apologize for this inadvertent error. We have now rectified the axis label, representing the starting weight at day 0 as 100% (currently Figure S1J).

4. In Figure 2, please indicate the database/s used for the identification of top biological pathways.

We used “WikiPathways subset of cellular processes” available at www.gsea-msigdb.org/gsea/msigdb/mouse/collections.jsp?targetSpeciesDB=Mouse#M8 for the pathway enrichment analysis presented in Figure 2B. We also utilized a previously published RA-target gene set for the gene set enrichment analysis presented in Figure 2C (Balmer and Blomhoff, 2002; [10.1194/jlr.r100015-jlr200](https://doi.org/10.1194/jlr.r100015-jlr200)). While this information was included in the materials and methods section in the original draft, we have now included these descriptions in the figure legend for further clarity. (please see revised figure legends 2B and 2C)

5. The authors show a more significant expansion in Tregs upon DSS treatment when non-canonical NF- κ B is ablated in DCs. Is this at the expense of a reduction of specific Th cells? Can the authors also report the number of cells beyond the % of cells?

In response to the reviewer’s comment, we have examined the abundance of Th17 cells in the colon of our knockout mice. As also observed earlier upon DC-specific ablation of NIK function (Jie et al., 2018), disruption of non-canonical NF- κ B signaling in DCs in *Relb*^{ADC} or *Nfkb2*^{ADC} mice led to a reduced frequency of RoR γ ⁺ Th17 cells in the LP (Figure S3F, new data). Our *in vitro* (Figure 2I) and *in vivo* (Figure 3F-G, 6B-6C) studies conclusively linked DC-intrinsic non-canonical NF- κ B signaling to intestinal Treg via the RA pathway. Therefore, we conjectured that the observed decline in the Th17 compartment in our knockouts could be secondary to Treg expansion. We have now further discussed this point in the revised manuscript. (line#296-300)

As the reviewer suggested, in addition to the Treg frequency, we have also presented the number of intestinal FoxP3⁺ CD4 T cells in the supplement (Figure S3E). Our data revealed a similar increase in the total Treg numbers in the mouse colon upon ablation of the non-canonical NF- κ B pathway in DCs. (line#296-300)

Revision Plan

6. In figure 6A, it appears that not only the amount of beta-catenin expressed but also the percentage of beta-catenin positive MNL DCs is significantly expanded upon ablation of non-canonical NFkb. Please verify and if so, include.

We thank the reviewer for this very insightful comment. We have now catalogued MLN DCs into β -catenin^{low} and β -catenin^{high} compartments. Indeed, we found a substantial more than two-fold increase in the frequency of β -catenin^{high} DCs in *Relb*^{ADC} mice. Accordingly, we have revised Figure 6A and emphasised this point in the text. (line#428-430)

7. In analogy to comment #1 above, please expand the analyses in human samples to include the expression of Relb and Nfkb2 to other myeloid cells.

Adhering to the valuable suggestion by the reviewer, we have now analysed the scRNAseq dataset (SCP 259) comparing DCs, macrophages, and inflammatory and cycling monocytes present in the human gut for the expression of RELB and NFKB2 mRNA (Figure 7B). Consistent with our observation involving the mouse colon, we found that mRNAs encoding these non-canonical signal transducers were mostly expressed in DCs among various MNPs. This point has also been emphasized in the revised draft. (line#446-448)

Reviewer #1 (Significance (Required)):

Strengths of the manuscript include the conceptual novelty of the intersection between non-canonical NFkb and the tolerogenic b-catenin-Raldh2 axis. And additional strength is the methodical approach, which includes various immunological and biochemical assessments as well as genetic perturbations to dissect such relationships. While it remains unknown the relevant triggers for the non-canonical axis described, this study advances our mechanistic understanding on how activation of this axis overrides regulatory mechanisms in DCs. As such, this manuscript should be of broad interest to immunologists and in particular mucosal immunologists.

We sincerely thank the reviewer for lauding our work as conceptually novel and methodical. The encouragement from the knowledgeable reviewer would certainly motivate us further to identify the relevant trigger of this pathway in the gut.

Reviewer #2 (Evidence, reproducibility and clarity (Required)):

The following issues are noted.

- all animal strains and their provenance should be described and properly referenced (for example, there are at least two CD11c-Cre strains with different specificity). Along the same lines, the specificity of Cre recombination should be confirmed, at least in major cell types (DCs vs T effector or regulatory cells).

We sincerely appreciate the reviewer's attention to these important details. We would like to point out that in our original draft, Table 1 in the Materials and Methods section provides information on the source and the identifier of all mouse strains used. In particular, we utilized CD11c-Cre mice with the identifier 008068 from the Jackson Laboratories. Alternately known as B6.Cg-Tg (Itgax-cre)1-1Reiz/J, this strain displays Cre-mediated recombination in more than 95% of conventional DCs while exhibiting only minor recombination in lymphocytes (<10%) and myeloid cells (<1%). This strain is also known not to increase recombination in CD11c^{low}-activated T cells (www.jax.org/strain/008068). Importantly, our immunoblot analyses revealed efficient depletion of RelB in specifically

Revision Plan

splenic CD11c⁺ cells of *Relb*^{ADC} mice with only a negligible reduction in CD11c⁻ cells (Figure S1E). Our analyses involving *Nfkb2*^{ADC} mice also assured of similar gene disruption specificity (Figure S1H). Notably, our results were consistent with those documented on *Relb*^{ADC} and *Nfkb2*^{ADC} strains earlier (Andreas et al., 2019). To further address the reviewer's concern pertaining to the T-cell compartment, we have now compared splenic CD4⁺ cells from *Nfkb2*^{fl/fl} and *Nfkb2*^{ADC} mice for the expression of p100 (Figure S1I, newly added in the revised draft). Our results confirmed that CD11c-Cre-driven ablation of the non-canonical NF-κB pathway did not perturb p100 expressions in T cells. Taken together, these allow us to emphasize that knockout phenotypes observed in our study were attributed to non-canonical NF-κB deficiency in DCs. We have accordingly modified the text to highlight gene deletion specificities in our knockouts. (line#166-167, 180-182)

- the DSS model is prone to "batch effects" of individual cages, and proper comparison between genotypes is possible only if mice of different genotypes (eg littermates) are housed together in the same cages. The authors should clearly confirm whether this was the case, and if not, key experiments should be repeated in this setting.

As mentioned in the materials and methods section of the original draft, littermate male mice of indicated genotypes were indeed cohoused for at least one week prior to experiments. We have now further emphasised this point in the legend of Figure 1.

- BMDCs represent a heterogeneous mixture of DCs and macrophages (Helft et al., Immunity 2015). These populations should be clearly defined and compared between genotypes, to make sure that they do not underlie the observed gene expression differences.

The knowledgeable reviewer has raised a very pertinent issue. We would like to emphasize that instead of generating BMDCs using GM-CSF alone following the protocol prescribed by Helft et al. (2015), we differentiated bone marrow cells to BMDCs using a cocktail of GM-CSF+IL4 adhering to the protocol published by Jin and Sprent (2018). Following the reviewer's suggestion, we have now compared BMDCs generated in these two protocols in our laboratory. As reported earlier (Jin and Sprent, 2018), unlike BMDCs generated using GM-CSF alone, BMDCs generated using the GM-CSF+IL4 cocktail did not contain CD115high macrophage-like cells (Figure S2C). (line#230-232) However, they displayed equivalent expressions of the DC marker CD135 on their surface. Moreover, when we compared BMDCs derived from *Relb*^{fl/fl} and *Relb*^{ADC11c} mice in flow cytometry analyses, we found comparable surface expression of CD135, assuring intact BMDC generation from bone marrow cells *ex vivo* in spite of the absence of RelB (Figure S2I). (line#266-268) These studies argue that macrophage-like cells did not contribute to the observed gene expression differences between WT and RelB-deficient BMDCs.

- the analysis of DCs in mutant strains (e.g. in Fig. 3) would benefit from a better definition of populations, e.g. resident vs migratory DCs in the MLN, the Notch2-dependent CD103+ CD11b+ DCs in the LP and MLN, etc. Again, this would be important to justify differences in gene expression (e.g. Fig. 3D).

We sincerely appreciate the comment from the knowledgeable reviewer. In a landmark paper from Prof. Fiona Powrie's group (Coombes et al., 2007), it was earlier demonstrated that CD103+ DCs present in the intestine migrate to local MLNs and play a key role in producing RA and supporting Tregs. While our BMDC data strongly supported a cell-intrinsic mechanism underlying Raldh2 upregulation upon non-canonical NF-κB deficiency (Figure 2F), our *in vivo* studies (Figure 3D-3E) did not entirely rule out also a possible expansion of RA-producing CD103+ DC compartment in our knockout mice. Although the

Revision Plan

proposition that non-canonical NF- κ B signaling regulates the generation of specific intestinal DC subsets seems attractive, we must point out that previous studies showed a relatively unaltered frequency of CD103+ cells among steady-state migratory DCs in skin-draining lymph nodes (Döhler et al., 2017). Nevertheless, following the reviewer's suggestion, we now plan to perform advanced flow cytometry analyses to compare *Relb*^{fl/fl} and *Relb* ^{Δ CD11c} mice for the frequency of CD103+CD11b-, CD103+CD11b+ and CD103-CD11b+ DCs in the intestine. To this end, we have already optimized our experimental protocol for staining intestinal DCs with anti-CD103 antibody (BD Bioscience). In the coming weeks, we are expecting to gather adequate numbers of littermate knockout mice to perform a side-by-side comparison. [NOTE: also the section - "Description of the planned revisions"]

- the analysis of b-catenin protein expression and cellular localization at the single-cell level (e.g. by IF) would greatly strengthen the mechanistic connection between NF- κ B and Wnt/b-catenin pathways.

Adhering to the reviewer's suggestion, we have now performed immunofluorescence assay (IFA) to capture the impact of RelB deficiency on β -catenin expression and cellular localization. Because BMDCs pose challenges for IFA owing to their non-adherent nature, we instead examined mouse embryonic fibroblasts (MEFs), which provide for a genetically amenable model cell system. As presented below (Figure R2), our IFA data conclusively demonstrated an increased cellular abundance and nuclear localization of β -catenin in *Relb*^{-/-} MEFs. While we are truly excited to find that our proposed mechanism is functional in another cell type, we feel that the inclusion of MEF data in the main manuscript, which describes DC-mediated immune controls, may cause significant distractions for the general

Figure R2: Immunofluorescence analysis revealing the basal expression of β -catenin in wild-type and *Relb*^{-/-} mouse embryonic fibroblasts. The bar plot (right) depicts the quantified levels of nuclear β -catenin in the indicated genotypes. The nuclei are stained with hoechst (blue) and β -catenin is detected as green. The panels represent independent fields of view. The mean fluorescence intensity of individual cells was determined using ImageJ. Fluorescence density analysis represent the average of three independent batches of cells (biological replicates, n =3). Scale bar = 20 μ m.

audience. Accordingly, we have provided this data for the reviewer's eyes only.

Revision Plan

Minor:

- The reanalysis of previous single-cell data in Figs. 1 and 7 are much less convincing or exciting than the new experimental data relegated to the supplements. The distribution of the results between main and experimental figures may be reconsidered in this light.

We concur with the knowledgeable reviewer that our scRNAseq analyses may have appeared less convincing in the original draft. In response to comments by reviewer-1 and reviewer-3, we have now added additional data panels (Figure 1B, Figure 7B and Figure 7G) and examined additional publicly available datasets (Figure S5). In the revised draft, these analyses helped us to more firmly establish a link between non-canonical NF- κ B signaling in DCs to aberrant intestinal inflammation in mice and humans.

However, we slightly diverge that many key experimental datasets were relegated to the supplement. Except for the FITC-dextran experiment, data from all other experimental analyses were presented in the main text (Figure 1). To suitably manage space in our figure panels, we opted to present quantified data averaged from experimental replicates in the main text while providing representative raw data in the supplement. Besides, immunoblot analyses confirming DC-specific ablation of target genes in our knockouts were placed in the supplement. Notably, these knockout strains were also examined earlier (Andreas et al., 2019). Those studies, along with our own analyses (Figure S1E, S1H and S1I - additional data), confirmed the most efficient gene deletion in CD11c+ cells. While maintaining these data in the supplement for want of space, we have now cited this reference in the main text to emphasize that knockout phenotypes observed in our study were attributed to non-canonical NF- κ B dysfunctions in DCs.

Reviewer #2 (Significance (Required)):

The manuscript by Deka et al. explores the role of the non-canonical NF- κ B pathway, specifically of its key mediators RelB and NF- κ B2, in dendritic cells (DCs) during intestinal inflammation. The key strength of the paper is the demonstration that DC-specific deletion of RelB or NF- κ B2 leads to improved acute or chronic DSS colitis. It is also shown that reducing the dose of b-catenin rescues the phenotype of RelB deletion, providing an important genetic connection between NF- κ B and Wnt/b-catenin pathways. As such, the work is novel, important and of potential significance to the field.

We express our deepest gratitude to the reviewer for his/her valuable time and insightful comments. We are indeed extremely excited that the knowledgeable reviewer finds our work novel, important and of potential significance to the field. These positive comments would inspire us to look further into potential interventions targeting the non-canonical NF- κ B pathway in human ailments.

Reviewer #3 (Evidence, reproducibility and clarity (Required)):

Summary:

This manuscript from Deka et al. investigates the role of dendritic cell noncanonical NF κ B signaling on intestinal inflammation. Based on prior data showing altered DC function in intestinal inflammation, they interrogated existing scRNAseq data and found that DSS treatment (which yields chemical colitis) increased the expression of non-canonical NF κ B family members in dendritic cells. This led to the generation of a DC-specific RelB deficient mouse and use of a DC specific NF κ B2 deficient mouse, each of which showed varying degrees of protection from chemical colitis.

Revision Plan

Overall, they do a very nice job identifying a mechanism by which noncanonical NF κ B signaling in dendritic cells contributes to intestinal inflammation via transcriptional regulation of Axin1, downregulation of β -catenin, restraint of Raldh2 synthesis, impaired retinoic acid synthesis and subsequent decrease in protective Tregs, IgA+ B cells, and microbial dysbiosis. The importance of this pathway is well supported by their focused targeting of β -catenin. After pharmacologic inhibition of β -catenin showed restoration of Raldh2 abundance, they made a DC-specific β -catenin haploinsufficiency RelB ^{Δ CD11c} mouse which showed impaired Raldh2 activity with restoration of colonic Tregs and fecal sIgA. When challenged with DSS, the protective phenotype seen with the RelB ^{Δ CD11c} was lost and the colitis phenotype returned to that of the Relb^{fl/fl} control, further solidifying the role of β -catenin, Raldh2 and RA on intestinal inflammation. Additionally, the discussion provides a robust mechanistic explanation for the phenotypic differences between the RelB and Nfkb2 genotypes, drawing on the authors' deep knowledge of the non-canonical NF κ B pathway.

Major comments:

1. Although they propose a novel mechanism by which dendritic cells can contribute to intestinal inflammation, it is in a model of acute epithelial injury that accentuates the contribution of the innate immune system. Would recommend including a discussion of the limitations of this model.

We most sincerely thank the knowledgeable reviewer for raising this important issue. We argue that while erosive epithelial injury initiates colitis in the DSS model, T cells were shown to aggravate intestinal pathologies, particularly at DSS doses used in our study (Kim et al., 2006; doi: 10.3748/wjg.v12.i2.302). Furthermore, our chemically-induced colitis model offered a convenient tool for genetically dissecting the DC-intrinsic role of the non-canonical NF- κ B pathway in the intestine. However, we agree entirely that no single animal model fully captures the clinical complexities of human IBD and that other models of experimental colitis should also be employed in the future to assess the generalisability of the proposed DC mechanism in regulating intestinal inflammation. In particular, future studies ought to examine composite knockout strains in the T-cell transfer model of experimental colitis to establish further the role of non-canonical NF- κ B signaling in DCs in alleviating intestinal inflammation. As suggested by the reviewer, we have now articulated this point in the discussion section. (line# 553-557)

2. The human work (Figure 7) shows solid evidence of heightened non-canonical NF κ B signaling in DCs via abundance of RELB and NFKB2 along with a few RelB important genes, however, the RA-specific pathway identified in the mouse work is not strongly corroborated by the human data. There is demonstration of one β -activated gene (CCND1) showing decreased expression in IBD patients, however no other gene along with RA pathway was clearly identified to be differentially expressed as one would predict from the mouse work.

We sincerely thank the knowledgeable reviewer for articulating this deficiency in our analyses of single-cell RNA-seq data derived from IBD patients (SCP259, Smillie et al., 2019). We would like to clarify that many well-known β -catenin target genes, including MYC, were not detectable in this dataset. Nevertheless, to address the reviewer's concern, we subjected this dataset to GSEA using a previously published list of RA target genes (Balmer et al., 2002). Our analyses revealed a significant enrichment of RA targets among genes that were downmodulated in DCs derived from inflamed colonic tissues of IBD patients as compared to those from non-inflamed tissues (Figure 7G, newly added in the revised

Revision Plan

version). We have now discussed this data in the result section. (line#470-475) These studies further substantiated the inverse correlation between noncanonical NF- κ B signalling and the RA pathway in DCs in the inflamed human gut.

Minor comments:

1. Their NF κ B2^{ACD11c} mouse underwent a regimen of chronic DSS treatment after acute DSS treatment only displayed subtle phenotypic changes. Was the same chronic colitis regimen also tested in the RelB^{ACD11c} ?

Indeed, we also examined *Relb*^{ACD11c} mice in the chronic DSS treatment regime. As compared to *Relb*^{fl/fl} mice, these knockout mice displayed significantly less bodyweight changes upon chronic DSS challenge. Because *Relb*^{ACD11c} mice readily showed acute DSS phenotype, we did not further pursue investigations involving this strain in the chronic DSS settings and rather focused on *Nfkb2*^{ACD11c} mice to illustrate chronic DSS phenotypes.

2. In the introduction, it was stated that patients with UC have a marked reduction in intestinal DCs. If DCs (particularly non-canonical NF κ B signaling) promote inflammation, how do you explain a decrease in this cell type in patients with active disease?

Depending on the expression of immunogenic or tolerogenic factors, DCs may both promote or subdue inflammation in the colon. We have now revisited the relevant reference published by Magnusson et al., (2016). Indeed, the authors noted a marked reduction in the intestine of the CD103+ DC subset, which has been majorly linked to tolerogenic RA synthesis. While it is generally thought that aberrant inflammation promotes the death of mononuclear phagocytes in the intestine, it seems that either a contraction of the tolerogenic DC compartment or downmodulation of tolerogenic pathways in DCs incites gut inflammation in IBD patients. We have now revised the text in the introduction section to clarify this point. (line#81-83)

3. The focus on retinoic acid is interesting, however may be oversimplifying the role of non-canonical NF κ B in DCs on the mucosal immune system. It must also be mentioned that there is crosstalk between the non-canonical and canonical NF κ B signaling systems, for example *Nfkb2* is capable of functioning as a I κ B protein and inhibiting RelA-p50 (from the last author's prior work - Basak et al, Cell, 2007). Thus would include some mention of possible effects on the canonical system that contribute to intestinal inflammation.

We thank the reviewer for raising this important point. As mentioned in the introduction section of our original draft, the canonical NF- κ B pathway in DCs aggravates experimental colitis mice (Visekruna, A. et al. 2015). Indeed, *Nfkb2*-dependent crosstalk was shown to modulate inflammatory RelA activity in a variety of cell types (Basak et al., 2007; Shih et al., 2009; Chawla et al., 2021). Although such cross-regulatory RelA controls by non-canonical NF- κ B signaling are yet to be established in DCs, our studies involving RelB-deficient cells confirmed an essential role of p100-mediated RelB regulations in DC functions. We admit that further studies are required to determine if, independent of RelB, p100 directs immunogenic DC attributes via also RelA or another factor. We have now elaborated on p100-mediated crosstalks in the discussion section. (line#561-562)

4. In the single-cell DSS data they analyzed, there was a distinct DC population seen with DSS colitis treatment. Although they are categorized as cDC2s, what genes separate them from the other DC populations?

We curated a list of genes from Brown et al., (2019) to categorize cDC1 and cDC2 subsets in our study. We would like to clarify that the list was provided in Supplementary

Revision Plan

Table 1 in our original draft. In view of the reviewer's comment, we have now referred to this Table in the legend of Supplementary Figure 1 and also in the main text. (line#153)

5. Why was the RNAseq work on BMDCs (that identified RA metabolism as a top-ranking differentially expressed pathway) done only on *Nfkb2*^{-/-} BMDCs and not *RelB*^{-/-}? *RelB*^{-/-} had a more pronounced protected phenotype in the cell type-specific knockout and is a cleaner target (does not have the IκB capability of *Nfkb2*).

We broadly agree with the knowledgeable reviewer that comparing WT and *Relb*^{-/-} BMDCs for global gene expressions could have been worthwhile. We would like to clarify that we initially utilized a dataset derived using *Nfkb2*^{-/-} BMDCs already available in the laboratory. These analyses were instrumental in developing a notion that non-canonical NF-κB signalling could be modulating *Raldh2* expression in DCs. Because previous studies involving germline *Relb*^{-/-} mice suggested a role of RelB in the nonhematopoietic niche in instructing myeloid development (Briseño et al., 2017), we focused our subsequent analyses on BMDCs generated using bone marrow cells from cell type-specific knockouts. Indeed, we could confirm elevated *Raldh2* expressions in BMDCs generated from both *Relb*^{ADC} and *Nfkb2*^{ADC} mice. Taken together, our studies suggested that *Nfkb2*-encoded p100 controlled *Raldh2* expressions in DCs by providing RelB:p52 and less so as a regulator of the RelA activity. Although we admit that further studies are required to determine if, independent of RelB, p100 directs immunogenic DC attributes via also RelA or another factor. We have now deliberated this point in the discussion section. (line#566-567)

Reviewer #3 (Significance (Required):

As a physician-scientist who clinically cares for patients with inflammatory bowel disease, and scientifically studies signaling within innate immune cells, this manuscript does a rigorous job of identifying a mechanism by which canonical NFκB signaling in dendritic cells contributes to intestinal inflammation. This study would be very informative for both basic and translational researchers as it identifies a clear pathway by which the innate immune system contributes to intestinal inflammation, and opens up room for inquiry into triggers of non-canonical NFκB in IBD and modulation of the RA pathway as a potential novel therapeutic target.

We are humbled that the knowledgeable reviewer finds our work to be informative for basic and translational research. These encouragements would undoubtedly motivate us further to identify the relevant trigger of this pathway in the gut and explore potential interventions.

4. Description of analyses that authors prefer not to carry out

None.

Dear Dr. Basak,

Thank you for submitting for consideration by The EMBO Journal your manuscript (EMBOJ-2024-117451-T) along with the reports of the three referees who evaluated it at Review Commons. I have now carefully read your manuscript, the referee comments, and your point-by-point response to them/revision plan, and I have also discussed them with the other members of our editorial team.

The referees are generally supportive of the study, they acknowledge the novelty, significance and relevance of the findings, and they point out that this work constitutes a significant contribution to the field. They also identify a few limitations, however, and they provide a number of suggestions for strengthening the work and the manuscript further.

Given the referees' positive comments and recommendations, I would like to invite you to submit a fully revised version of your manuscript, addressing the comments of all three reviewers along the lines described in your point-by-point response. I would like to note that -in line with the referee comments- we think that the identification of the trigger and mechanism (or, at least, part of it) that leads to high levels of non-canonical NF- κ B signaling in the gut would strengthen the manuscript considerably, and we therefore encourage you to expand your investigation in this direction. Please also include in your re-submission a detailed point-by-point response to the referees' comments explaining all changes to the manuscript.

I should add that it is EMBO Journal policy to allow only a single round of major revision, and acceptance of your manuscript will therefore depend on the completeness of your responses in this revised version. If you have any questions or comments, we can also discuss the revisions in a video chat, if you like.

We generally allow three months as standard revision time (14th July 2024). As a matter of policy, competing manuscripts published during this period will not negatively impact our assessment of the conceptual advance presented by your study. However, we request that you contact us as soon as possible upon publication of any related work, to discuss how to proceed. Should you foresee a problem in meeting this three-month deadline, please let us know in advance and we may be able to grant an extension.

Thank you for the opportunity to consider your work for publication in The EMBO Journal. I look forward to your revision.

Yours sincerely,

Instructions for preparing your revised manuscript

1. When you are ready to submit the revision, please upload:

- A Word file of the manuscript text (including legends of main Figures, EV Figures and Tables). Please make sure that changes are highlighted (or "tracked") to be clearly visible.

- Individual production-quality figure files (one file per figure). When assembling your figures, please refer to our figure preparation guidelines in order to ensure proper formatting and readability in print as well as on screen:

If the data shown in a figure are obtained from n {less than or equal to} 2, please use scatter plots showing the individual data points.

- i. the name of the statistical test used to generate error bars and P values
- ii. the number (n) of independent experiments (please specify technical or biological replicates) underlying each data point (discussion of statistical methodology can be reported in the Materials and Methods section, but figure legends should contain a basic description of n , P , and the test applied)
- iii. the nature of the bars and error bars (s.d., s.e.m.).

- A point-by-point response to the referees' comments, with a detailed description of the changes made (as a word file). All

referees' concerns must be fully addressed and their suggestions taken on board. When preparing your letter of response to the referees' comments, please bear in mind that this will form part of the Review Process File and will therefore be available online to the community. Please note that you have the possibility to opt out of the transparent process at any stage prior to publication by letting the editorial office know (contact@embojournal.org); if you do opt out, the Review Process File link will point to the following statement: "No Review Process File is available with this article, as the authors have chosen not to make the review process public in this case.". For more details on our Transparent Editorial Process, please visit our website: <https://www.embopress.org/page/journal/14602075/authorguide#transparentprocess>

- Expanded View (EV) files (replacing Supplementary Information) that are collapsible/expandable online. A maximum of 5 EV Figures can be typeset. EV Figures should be cited as "Figure EV1, Figure EV2" etc. in the text, and their respective legends should be included in the manuscript file after the legends of regular figures. See detailed instructions regarding Expanded View files here:

- For the figures that you do NOT wish to display as Expanded View figures, they should be bundled together with their legends in a single PDF file called "Appendix", which should start with a short Table of Contents (including page numbers). Appendix figures should be referred to in the main text as: "Appendix Figure S1, Appendix Figure S2" etc. Please see detailed instructions here: <https://www.embopress.org/page/journal/14602075/authorguide#expandedview>

- Please note that you can use our optional "Structured Methods" format for your Materials and Methods. This includes a "Reagents and Tools" Table listing all key reagents, equipment and software used in the study, followed by a "Methods and Protocols" section, where your methods can be described in detail. Please see detailed instructions, examples and templates in our author guide: <https://www.embopress.org/page/journal/14602075/authorguide#researcharticleguide>

- A complete author checklist, which you can download from our author guidelines (<https://www.embopress.org/page/journal/14602075/authorguide>). Please note that the checklist will also be part of the Review Process File.

2. Please note that no statistics should be calculated if $n=2$.

3. Before submitting your revision, primary datasets (and computer code, where appropriate) produced in this study need to be deposited in appropriate public databases (see <https://www.embopress.org/page/journal/14602075/authorguide#dataavailability>).

The accession numbers and database should be listed in a formal "Data availability" section (placed after Materials and Methods) that follows the model below (see also

<https://www.embopress.org/page/journal/14602075/authorguide#dataavailability>):

Data availability

- RNA-seq data: Gene Expression Omnibus GSE46843 (<https://www.ncbi.nlm.nih.gov/geo/query/acc.cgi?acc=GSE46843>)
- [data type]: [name of the resource] [accession number/identifier/doi] ([URL or identifiers.org/DATABASE:ACCESSION])

*** Note: all links should resolve to a page where the data can be accessed. ***

*** Note: the Data Availability Section is restricted to new primary data that are part of this study. ***

4. Please check that the title and the abstract of the manuscript are brief, yet explicit, even to non-specialists. The length of the title should not exceed 100 characters (including spaces), and the abstract should be a single paragraph not exceeding 175 words.

5. Please also note our reference format: <https://www.embopress.org/page/journal/14602075/authorguide#referencesformat>.

7. Please remember: digital image enhancement is acceptable practice, as long as it accurately represents the original data and conforms to community standards. If a figure has been subjected to significant electronic manipulation, this must be noted in the figure legend or in the "Materials and Methods" section. The editors reserve the right to request original versions of figures and the original images that were used to assemble the figure.

8. Our journal encourages inclusion of data citations in the reference list to directly cite datasets that were obtained from public databases. Data citations in the article text are distinct from normal bibliographical citations and should directly link to the database records from which the data can be accessed. In the main text, data citations are formatted as follows: "Data ref: Smith et al, 2001" or "Data ref: NCBI Sequence Read Archive PRJNA342805, 2017". In the Reference list, data citations must be labeled with "[DATASET]". A data reference must provide the database name, accession number/identifiers, and a resolvable link to the landing page from which the data can be accessed at the end of the reference. Further instructions are available at: <https://www.embopress.org/page/journal/14602075/authorguide#referencesformat>.

9. We request authors to consider both actual and perceived competing interests. Please review our policy (<https://www.embopress.org/page/journal/14602075/authorguide#conflictsofinterest>) and update your competing interests statement if necessary. Please name this section 'Disclosure and competing interests statement' and place it after the Acknowledgements section.

10. Please note that all corresponding authors are required to provide an ORCID ID upon submission of a revised manuscript (<https://orcid.org/>). Please find instructions on how to link your ORCID ID to your account in our manuscript tracking system in our Author guidelines (<https://www.embopress.org/page/journal/14602075/authorguide#authorshipguidelines>).

11. We use CRediT to specify the contributions of each author in the journal submission system. CRediT replaces the author contribution section, which should be removed from the manuscript. Please use the free text box to provide more detailed descriptions. See also guide to authors: <https://www.embopress.org/page/journal/14602075/authorguide#authorshipguidelines>.

13. We would also welcome the submission of cover suggestions or motifs to be used by our Graphics Illustrator in designing a cover.

14. Please use the link below to submit your revision:
Link Not Available

Rev_Com_number: RC-2024-02368

New_manu_number: EMBOJ-2024-117451-T

Corr_author: Basak

Title: Non-canonical NF- κ B signaling promotes gut inflammation by restraining the tolerogenic b-catenin-Raldh2 axis in dendritic cells

Point-by-point response to Reviewers' comments:**Reviewer #1 (Evidence, reproducibility and clarity (Required)):**

Deka and colleagues report that a non-canonical NF κ B signaling operates in DCs in the context of inflammation and inhibits a tolerogenic mechanism driven by b-catenin-Raldh2. The following comments are made to clarify the findings presented.

1. The authors re-analyzed published scRNAseq data from DSS colitis to identify the expression of *Relb* and *NFKB2* in myeloid cells.

a. The authors are encouraged to expand this analysis to other published datasets.

We sincerely appreciate the comment from the knowledgeable reviewer. Unfortunately, we did not find any other publicly available scRNAseq dataset from DSS-treated mice. To circumvent this problem, we instead examined a previously published microarray-based bulk transcriptomic dataset obtained using FACS-sorted DCs isolated from the mouse colon (GSE58446, Muzaki et al. 2016; doi:10.1038/mi.2015.64). We consistently found an increased expression of *Relb*, and to some extent *Nfkb2*, mRNAs in intestinal DCs upon DSS treatment (Fig R1). Because microarray analysis lacks the quantitative attributes that scRNAseq offers, we provided this newly analysed dataset for reviewer's eyes and refrained from including this data in the manuscript *per se*.

Importantly, we could identify an additional scRNAseq dataset derived human ulcerative colitis patients (GSE162335, Devlin et al. 2021, doi.org/10.1053/j.gastro.2020.12.030). Interrogation of this dataset indeed confirmed that *RELB* and *NFKB2* mRNAs were majorly expressed in intestinal DCs and not in intestinal macrophages and that IBD was associated with increased expression of multiple *RelB*-important genes in DCs. These analyses further supported the notion that heightened non-canonical NF- κ B signalling in DCs could be fuelling aberrant gut inflammation. We have now incorporated this newly acquired data in the Appendix Fig. S5B-S5D. (discussed in line #411-417)

b. Additionally, the expression of *Relb* and *NFKB2* in other cells - especially other myeloid cells - should be explored and included, even if the authors later choose to test their function in DCs.

Adhering to this brilliant suggestion, we have now interrogated the mouse scRNAseq dataset (GSE148794; Ho et al., 2021) to compare macrophages and DCs for the expression of the non-canonical signal transducers. Indeed, we found a relatively insignificant level of *Relb* and *Nfkb2* mRNAs in intestinal macrophages in comparison to intestinal DCs (Fig. 1B). (discussed in line #137-140) Our data suggested that the non-canonical NF- κ B pathway is likely to play a more prominent role in DCs than in macrophages in the gut. This comparison proved useful in motivating subsequent in-depth analyses of this pathway in DCs in the context of experimental colitis.

c. Please note that there is a transition from Fig 1A to Fig 1B to focus on DCs, which is not apparent from the figure.

Please find our response to #1b.

2. Please include scale bars for all histological analyses.

We thank the reviewer for alerting us. The scale bars were already included in the histological analyses; we have now appropriately highlighted them for better visual clarity.

3. In Fig S1I, the authors show that loss of body weight upon DSS treatment in *Nfkb2*^{ACD11c} is indistinguishable from control. Why is the starting weight at 110%? Please clarify.

We sincerely apologize for this inadvertent error. We have now rectified the axis label, representing the starting weight at day 0 as 100% (currently Appendix Fig. S1K).

4. In Figure 2, indicate the database/s used for the identification of top biological pathways.

We used “WikiPathways subset of cellular processes” available at www.gsea-msigdb.org/gsea/msigdb/mouse/collections.jsp?targetSpeciesDB=Mouse#M8 for the pathway enrichment analysis (Fig. 2B). We also utilized a previously published RA-target gene set for the gene set enrichment analysis (Fig. 2C) (Balmer and Blomhoff, 2002; [10.1194/jlr.r100015-jlr200](https://doi.org/10.1194/jlr.r100015-jlr200)). While this information was included in the Materials and Methods section in the original draft, we have now included these descriptions also in the figure legend for further clarity.

5. The authors show a more significant expansion in Tregs upon DSS treatment when non-canonical NFκB is ablated in DCs. Is this at the expense of a reduction of specific Th cells? Can the authors also report the number of cells beyond the % of cells?

In response to the reviewer’s comment, we have now examined the abundance of intestinal Th17 cells. As also observed earlier upon DC-specific ablation of NIK (Jie et al., 2018), disruption of non-canonical NF-κB signaling in DCs in *Relb^{ADC}* or *Nfkb2^{ADC}* mice led to a reduced frequency of RoRγt⁺ Th17 cells in the LP (Appendix Fig. S3G, new data). On the other hand, our *in vitro* (Fig. 2I) and *in vivo* (Fig. 3F-3G, 6B-6C) studies conclusively linked DC-intrinsic non-canonical NF-κB signaling to intestinal Tregs via the RA pathway. Therefore, we conjectured that the observed decline in the Th17 compartment in our knockouts could be secondary to Treg expansion. We have now further discussed this point in the draft. (line #290-292)

Next, in addition to the Treg frequency, we have also presented the total number of intestinal FoxP3+ CD4 T cells (Appendix Fig. S3F). Our data revealed a similar increase in Treg numbers in the mouse colon upon ablation of the non-canonical NF-κB pathway in DCs.

6. In figure 6A, it appears that not only the amount of beta-catenin expressed but also the percentage of beta-catenin positive MNL DCs is significantly expanded upon ablation of non-canonical NFκB. Please verify and if so, include.

We thank the reviewer for this insightful comment. We have now catalogued MLN DCs into β-catenin^{low} and β-catenin^{high} compartments. Indeed, we found a more than two-fold increase in the frequency of intestinal β-catenin^{high} DCs in *Relb^{ADC}* mice. Accordingly, we have revised Figure 6A and emphasised this point in the text. (line #378-379)

7. In analogy to comment #1 above, please expand the analyses in human samples to include the expression of Relb and Nfkb2 to other myeloid cells.

Adhering to the valuable suggestion by the reviewer, we have now further analysed the scRNAseq dataset (SCP 259) to compare the abundance of RELB and NFKB2 mRNAs in human DCs and other myeloid cells (current Fig. 7B). Also, consistent with our observation involving the mouse colon, we found that mRNAs encoding RELB, NFKB and LTBR were almost equivalently expressed by intestinal DCs in the human gut. This point has also been emphasized in the revised draft. (line #401-402, line#406-407)

Reviewer #1 (Significance (Required)):

Strengths of the manuscript include the conceptual novelty of the intersection between non-canonical NFκB and the tolerogenic β-catenin-Raldh2 axis. And additional strength is the methodical approach, which includes various immunological and biochemical assessments as well as genetic perturbations to dissect such relationships. While it remains unknown the relevant triggers for the non-canonical axis described, this study advances our mechanistic understanding on how activation of this axis overrides regulatory mechanisms in DCs. As such, this manuscript

should be of broad interest to immunologists and in particular mucosal immunologists.

We sincerely thank the reviewer for these encouraging comments and his/her valuable time. In response to the reviewer's comments and suggestions from the Editor, we have now expanded the scope of our study to investigate the potential trigger of the non-canonical NF- κ B pathway in intestinal DCs.

As such, LT β R and GM-CSFR both have been implicated in the non-canonical NF- κ B activation in DCs (doi: 10.1016/j.immuni.2014.02.006; doi: 10.4049/jimmunol.2000197). Our single-cell RNA-seq data analyses revealed a substantial expression of mRNAs encoding LT β R and GM-CSFR, alternately termed Csf2R, in intestinal DCs even from mice that were not challenged (Fig. 1C). A significant fraction of intestinal lymphocytes expressed the lymphotoxin ligand *Ltb*, whose abundance was elevated upon DSS treatment (Fig. 1F). However, immune cells expressing *Csf2*, the cognate ligand for GM-CSFR, were less prevalent in the gut. Accordingly, we reasoned that LT β R stimulated non-canonical NF- κ B signaling in intestinal DCs. Indeed, disruption of lymphotoxin-mediated LT β R engagement in mice using LT β R-IgG fusion protein downmodulated the non-canonical NF- κ B pathway in intestinal DCs (Fig. 1E, Appendix Fig. S1E). Finally, DCs present also in the human gut, particularly in IBD patients, proficiently expressed LT β R mRNA (Fig. 7D). Previous studies suggested that LT β R modulates intestinal DC functions (doi: 10.1016/j.chom.2011.06.002). Our analyses placed the non-canonical NF- κ B pathway downstream of LT β R in these DCs. Admittedly, additional genetic studies will be important to fully dissect the molecular triggers of this pathway in various intestinal cell subsets.

Reviewer #2 (Evidence, reproducibility and clarity (Required)):

The following issues are noted.

- all animal strains and their provenance should be described and properly referenced (for example, there are at least two CD11c-Cre strains with different specificity). Along the same lines, the specificity of Cre recombination should be confirmed, at least in major cell types (DCs vs T effector or regulatory cells).

We sincerely appreciate the reviewer's attention to these important details. We would like to point out that in our original draft, Table 1 in the Materials and Methods section provided information on the source and the identifier of all the mouse strains. In particular, we utilized CD11c-Cre mice with the identifier 008068 from the Jackson Laboratories. Alternately known as B6.Cg-Tg (*Itgax-cre*)1-1Reiz/J, this strain displays Cre-mediated recombination in more than 95% of conventional DCs while exhibiting only minor recombination in lymphocytes and myeloid cells. This strain is also known not to increase recombination in CD11c^{low}-activated T cells. Importantly, our immunoblot analyses revealed efficient depletion of RelB in specifically splenic CD11c⁺ cells from *Relb*^{ADC} mice with only a negligible reduction in CD11c⁻ cells (Appendix Fig. S1F). Our analyses involving *Nfkb2*^{ADC} mice also assured of similar gene disruption specificity (Appendix Fig. S1I). Our results were consistent with those documented on *Relb*^{ADC} and *Nfkb2*^{ADC} strains earlier (Andreas et al., 2019). To further address the reviewer's concern, we have now compared splenic CD4⁺ cells from *Nfkb2*^{fl/fl} and *Nfkb2*^{ADC} mice for the expression of p100 (Appendix Fig. S1J). Our results confirmed that CD11c-Cre-driven ablation of the non-canonical NF- κ B pathway did not perturb p100 expressions in T cells. Taken together, these allow us to emphasize that knockout phenotypes observed in our study were attributed to non-canonical NF- κ B deficiency in DCs. We have accordingly modified the text to highlight gene deletion specificities in our knockouts. (line#182-183, 197-199)

- the DSS model is prone to "batch effects" of individual cages, and proper comparison between genotypes is possible only if mice of different genotypes (eg littermates) are housed together in the same cages. The authors should clearly confirm whether this was the case, and if not, key experiments should be repeated in this setting.

As mentioned in the Materials and Methods section, littermate male mice of the indicated genotypes were cohoused for at least one week prior to experiments. We have now highlighted this point in the legend of Figure 1.

- BMDCs represent a heterogeneous mixture of DCs and macrophages (Helft et al., Immunity 2015). These populations should be clearly defined and compared between genotypes, to make sure that they do not underlie the observed gene expression differences.

The knowledgeable reviewer has raised a very pertinent issue. We would like to point out that instead of generating BMDCs using GM-CSF alone following the protocol by Helft et al., (2015), we differentiated bone marrow cells to BMDCs using a cocktail of GM-CSF+IL4 adhering to the protocol published by Jin and Sprent (2018). Following the reviewer's suggestion, we have now compared BMDCs generated in these two protocols in our laboratory. As reported earlier (Jin and Sprent, 2018), BMDCs generated using GM-CSF+IL4 did not contain CD115-high macrophage-like cells unlike those generated using GM-CSF alone (Appendix Fig. S2C). (line #219-221) However, these sets displayed equivalent expressions of the DC marker CD135 on their surface. Furthermore, BMDCs derived from *Relb^{fl/fl}* and *Relb^{ACD11c}* mice displayed comparable surface expression of CD135, assuring intact DC differentiation from bone marrow cells *ex vivo* in spite of the absence of RelB (Appendix Fig. S2I). These studies argue that macrophage-like cells did not contribute to the observed gene expression differences between WT and RelB-deficient BMDCs.

- the analysis of DCs in mutant strains (e.g. in Fig. 3) would benefit from a better definition of populations, e.g. resident vs migratory DCs in the MLN, the Notch2-dependent CD103+ CD11b+ DCs in the LP and MLN, etc. Again, this would be important to justify differences in gene expression (e.g. Fig. 3D).

We sincerely appreciate these comments from the knowledgeable reviewer. In a landmark paper from Prof. Fiona Powrie's group (Coombes *et al.*, 2007), it was demonstrated that CD103+ DCs present in the intestine migrate to local MLNs and play a key role in producing RA. While our BMDC data strongly supported a cell-intrinsic mechanism underlying Raldh2 upregulation upon non-canonical NF- κ B deficiency (Fig. 2F), our *in vivo* studies (Fig. 3D-3E) did not entirely rule out a possible expansion of this RA-producing CD103+ DC compartment in our knockout mice. We have now performed flow cytometry analyses to compare *Relb^{fl/fl}* and *Relb^{ACD11c}* mice for the frequency of CD103+CD11b-, CD103+CD11b+ and CD103-CD11b+ DCs in the gut. Our studies assured only a minor change in the frequency of intestinal CD103+ DCs in our knockout mice (Appendix Fig. S3C). Our results were consistent to a previous report revealing a relatively unaltered frequency of CD103+ cells among steady-state migratory DCs in skin-draining lymph nodes of *Relb^{ACD11c}* mice (Döhler et al., 2017, doi: 10.3389/fimmu.2017.00726). Taken together, we conclude that the colitogenic phenotype observed in our knockout mice were not due to an altered frequency of intestinal DC subsets. (line #268-269)

- the analysis of b-catenin protein expression and cellular localization at the single-cell level (e.g. by IF) would greatly strengthen the mechanistic connection between NF- κ B and Wnt/b-catenin pathways.

Adhering to the reviewer's suggestion, we have performed immunofluorescence assay (IFA) to capture the impact of RelB deficiency on β -catenin expression and cellular localization. Because BMDCs pose challenges for IFA owing to their non-adherent nature, we instead examined mouse embryonic fibroblasts (MEFs), which provide for a genetically amenable model cell system. As presented below (Fig. R2), our IFA data conclusively demonstrated an increased

cellular abundance and nuclear localization of β -catenin in *Relb*^{-/-} MEFs. While we are truly excited to find that our proposed mechanism is functional in another cell type, we feel that the inclusion of MEF data in the main manuscript, which describes DC-mediated immune controls, may cause distraction for the general audience. Accordingly, we have provided this data for the reviewer's eyes only.

Minor:

- The reanalysis of previous single-cell data in Figs. 1 and 7 are much less convincing or exciting than the new experimental data relegated to the supplements. The distribution of the results between main and experimental figures may be reconsidered in this light.

We concur that our single-cell RNA-seq analyses may have appeared less convincing in the original draft. In response to comments by reviewer-1 and reviewer-3, we have added additional data panels (Fig. 1B, Fig. 7B, Fig. 7D and Fig. 7F and examined additional publicly available datasets (Appendix Fig. S5B-D). These analyses more firmly establish a link between non-canonical NF- κ B signaling in DCs to aberrant intestinal inflammation in mice and humans.

However, we slightly diverge that many key experimental datasets were relegated to the supplement. Except for the FITC-dextran experiment (Appendix Fig. S1H), data from all other experimental analyses were presented in the main text (Fig. 1). To suitably manage space in our figure panels, we opted to present quantified data averaged from experimental replicates in the main text while providing representative raw data in the supplement. Besides, immunoblot analyses confirming DC-specific ablation of target genes in our knockouts were placed in the supplement (Appendix Fig. S1F, S1I and S1J). Notably, these strains were also examined earlier (doi: 10.4049/jimmunol.1801530). While maintaining these data in the supplement for the want of space, we have now cited this reference in the main text to emphasize that knockout phenotypes observed in our study were attributed to non-canonical NF- κ B dysfunctions specifically in DCs. (line #182-183)

Reviewer #2 (Significance (Required)):

The manuscript by Deka et al. explores the role of the non-canonical NF- κ B pathway, specifically of its key mediators RelB and NF- κ B2, in dendritic cells (DCs) during intestinal inflammation. The key strength of the paper is the demonstration that DC-specific deletion of RelB or NF- κ B2 leads to improved acute or chronic DSS colitis. It is also shown that reducing the dose of β -catenin rescues the phenotype of RelB deletion, providing an important genetic connection between NF- κ B and Wnt/ β -catenin pathways. As such, the work is novel, important and of potential significance to the field.

We express our deepest gratitude to the knowledgeable reviewer for his/her valuable time and insightful comments. We are indeed excited that the reviewer finds our work novel, important and of potential significance.

Reviewer #3 (Evidence, reproducibility and clarity (Required)):

Summary:

This manuscript from Deka et al. investigates the role of dendritic cell noncanonical NF κ B signaling on intestinal inflammation. Based on prior data showing altered DC function in intestinal inflammation, they interrogated existing scRNAseq data and found that DSS treatment (which yields chemical colitis) increased the expression of non-canonical NF κ B family members in dendritic cells. This led to the generation of a DC-specific RelB deficient mouse and use of a DC specific NF κ B2 deficient mouse, each of which showed varying degrees of protection from chemical colitis.

Overall, they do a very nice job identifying a mechanism by which noncanonical NF κ B signaling in dendritic cells contributes to intestinal inflammation via transcriptional regulation of Axin1, downregulation of β -catenin, restraint of Raldh2 synthesis, impaired retinoic acid synthesis and subsequent decrease in protective Tregs, IgA+ B cells, and microbial dysbiosis. The importance of this pathway is well supported by their focused targeting of β -catenin. After pharmacologic inhibition of β -catenin showed restoration of Raldh2 abundance, they made a DC-specific β -catenin haploinsufficiency RelB^{ACD11c} mouse which showed impaired Raldh2 activity with restoration of colonic Tregs and fecal sIgA. When challenged with DSS, the protective phenotype seen with the RelB^{ACD11c} was lost and the colitis phenotype returned to that of the Relb^{fl/fl} control, further solidifying the role of β -catenin, Raldh2 and RA on intestinal inflammation. Additionally, the discussion provides a robust mechanistic explanation for the phenotypic differences between the RelB and Nfkb2 genotypes, drawing on the authors' deep knowledge of the non-canonical NF κ B pathway.

Major comments:

1. Although they propose a novel mechanism by which dendritic cells can contribute to intestinal inflammation, it is in a model of acute epithelial injury that accentuates the contribution of the innate immune system. Would recommend including a discussion of the limitations of this model.

This is an important issue. We argue that while epithelial injury initiates colitis in the DSS model, T cells were shown to aggravate intestinal pathologies, particularly at DSS doses used in our study (Kim et al., 2006; doi: 10.3748/wjg.v12.i2.302). Furthermore, our chemically-induced colitis model offered a convenient tool for genetically dissecting the DC-intrinsic role of the non-canonical NF- κ B pathway in the intestine. However, we agree that no single animal model fully captures the clinical complexities of human IBD and that other models of experimental colitis should also be employed to assess the generalisability of the proposed DC mechanism in regulating intestinal inflammation. In particular, future studies ought to examine composite knockout strains in the T-cell transfer model to establish further the role of non-canonical NF- κ B signaling in DCs in alleviating intestinal inflammation. As suggested by the reviewer, we have now articulated this point in the discussion section. (line #527-530)

2. The human work (Figure 7) shows solid evidence of heightened non-canonical NF κ B signaling in DCs via abundance of RELB and NFKB2 along with a few RelB important genes, however, the RA-specific pathway identified in the mouse work is not strongly corroborated by the human data. There is demonstration of one β -activated gene (CCND1) showing decreased expression in IBD patients, however no other gene along with RA pathway was clearly identified to be differentially expressed as one would predict from the mouse work.

We sincerely thank the knowledgeable reviewer for articulating this deficiency in our analyses of single-cell RNA-seq data derived from IBD patients (SCP259, Smillie et al., 2019). We would like to clarify that many well-known β -catenin target genes, including MYC, were not detectable in this dataset. Nevertheless, to address the reviewer's concern, we subjected this dataset to GSEA using a previously published list of RA target genes (Balmer et al., 2002). Our analyses revealed a significant enrichment of RA targets among genes that were downmodulated in DCs derived from inflamed colonic tissues of IBD patients as compared to those from non-inflamed tissues (Fig. 7F). We have now discussed this data in the result section. (line #426-432) These studies substantiated the inverse correlation between noncanonical NF- κ B signalling and the RA pathway in DCs in the inflamed human gut.

Minor comments:

1. Their NF κ B2^{ACD11c} mouse underwent a regimen of chronic DSS treatment after acute DSS treatment only displayed subtle phenotypic changes. Was the same chronic colitis regimen also tested in the RelB^{ACD11c}?

Indeed, we also examined *Relb*^{ACD11c} mice in the chronic DSS regime. As compared to *Relb*^{fl/fl} mice, these knockout mice displayed significantly less bodyweight changes upon chronic DSS challenge. Because *Relb*^{ACD11c} mice readily showed acute DSS phenotype, we did not further pursue investigations involving this strain in the chronic DSS settings and rather focused on *Nfkb2*^{ACD11c} mice to illustrate chronic DSS phenotypes.

2. In the introduction, it was stated that patients with UC have a marked reduction in intestinal DCs. If DCs (particularly non-canonical NFκB signaling) promote inflammation, how do you explain a decrease in this cell type in patients with active disease?

Depending on the expression of immunogenic or tolerogenic factors, DCs may both promote or subdue inflammation in the colon. We have now revisited the relevant reference published by Magnusson et al., (2016). Indeed, the authors noted a marked reduction in the intestine of the CD103+ DC subset, which has been majorly linked to tolerogenic RA synthesis. While it is generally thought that aberrant inflammation promotes the death of mononuclear phagocytes in the intestine, it seems that either a contraction of the tolerogenic DC compartment or downmodulation of tolerogenic pathways in DCs incites gut inflammation in IBD patients. We have now revised the text in the introduction section to clarify this point. (line #78-79)

3. The focus on retinoic acid is interesting, however may be oversimplifying the role of non-canonical NFκB in DCs on the mucosal immune system. It must also be mentioned that there is crosstalk between the non-canonical and canonical NFκB signaling systems, for example *Nfkb2* is capable of functioning as a IκB protein and inhibiting RelA-p50 (from the last author's prior work - Basak et al, Cell, 2007). Thus would include some mention of possible effects on the canonical system that contribute to intestinal inflammation.

We thank the reviewer for raising this important point. As mentioned in the introduction section of our original draft, the canonical NF-κB pathway in DCs aggravates experimental colitis in mice (Visekruna, A. et al. 2015). *Nfkb2*-encoded p100 not only controls RelB, but also RelA. Indeed, *Nfkb2*-dependent crosstalk was shown to modulate inflammatory RelA activity in a variety of cell types (Basak et al., 2007; Shih et al., 2009; Chawla et al., 2021). Although such cross-regulatory RelA controls by non-canonical NF-κB signaling are yet to be established in DCs, our studies involving RelB-deficient cells confirmed an essential role of p100-mediated RelB regulations in DC functions. We admit that further studies are required to determine if, independent of RelB, p100 directs immunogenic DC attributes via also RelA or another factor. We have now elaborated on p100-mediated crosstalks in the discussion section. (line #534-537)

4. In the single-cell DSS data they analyzed, there was a distinct DC population seen with DSS colitis treatment. Although they are categorized as cDC2s, what genes separate them from the other DC populations?

We curated a list of genes from Brown et al., (2019) to categorize cDC1 and cDC2 subsets in our study. We would like to clarify that the list was provided in Appendix Table S1 in our original draft. In view of the reviewer's comment, we have now referred to this Table in the legend of Appendix Figure S1 and also in the main text. (line #154)

5. Why was the RNAseq work on BMDCs (that identified RA metabolism as a top-ranking differentially expressed pathway) done only on *Nfkb2*^{-/-} BMDCs and not *RelB*^{-/-}? *RelB*^{-/-} had a more pronounced protected phenotype in the cell type-specific knockout and is a cleaner target (does not have the IκB capability of *Nfkb2*).

We broadly agree that comparing WT and *Relb*^{-/-} BMDCs in transcriptomics analyses could have been worthwhile. We would like to clarify that we initially utilized a dataset already available in the laboratory but obtained using BMDCs derived from global *Nfkb2*^{-/-} mice. These analyses were instrumental in developing the notion that non-canonical NF-κB signalling could be modulating *Radh2* expression in DCs. Because previous studies involving germline *Relb*^{-/-}

mice suggested a role of RelB in the nonhematopoietic niche in instructing hematopoietic cell development (Briseño et al., 2017), we did not pursue transcriptomic studies using BMDCs from these mice. We rather focused our subsequent analyses on BMDCs generated using bone marrow cells from cell type-specific knockouts. Indeed, we could confirm elevated *Raldh2* expressions in BMDCs generated from both *Relb*^{ADC} and *Nfkb2*^{ADC} mice. Taken together, our studies suggested that *Nfkb2*-encoded p100 controlled *Raldh2* expressions in DCs by providing RelB:p52 and less so as a regulator of the RelA activity. Although we admit that further studies are required to determine if, independent of RelB, p100 directs immunogenic DC attributes via also RelA or another factor. We have now deliberated this point in the discussion section. (line #537-540)

Reviewer #3 (Significance (Required):

As a physician-scientist who clinically cares for patients with inflammatory bowel disease, and scientifically studies signaling within innate immune cells, this manuscript does a rigorous job of identifying a mechanism by which canonical NFκB signaling in dendritic cells contributes to intestinal inflammation. This study would be very informative for both basic and translational researchers as it identifies a clear pathway by which the innate immune system contributes to intestinal inflammation, and opens up room for inquiry into triggers of non-canonical NFκB in IBD and modulation of the RA pathway as a potential novel therapeutic target.

We are humbled that the knowledgeable reviewer finds our work to be informative for basic and translational research. We also sincerely thank the reviewer for his/her valuable time. In response to the reviewer's comments and suggestions from the Editor, we have expanded the scope of our study to investigate the potential trigger of the non-canonical NF-κB pathway.

As such, LTβR and GM-CSFR both have been implicated in the non-canonical NF-κB activation in DCs (doi: 10.1016/j.immuni.2014.02.006; doi: 10.4049/jimmunol.2000197). Our single-cell RNA-seq data analyses revealed a substantial expression of mRNAs encoding LTβR and GM-CSFR, alternately termed *Csf2R*, in intestinal DCs even from mice that were not challenged (Fig. 1C). A significant fraction of intestinal lymphocytes expressed the lymphotoxin ligand *Ltb*, whose abundance was elevated upon DSS treatment (Fig. 1F). However, immune cells expressing *Csf2*, the cognate ligand for GM-CSFR, were less prevalent in the gut. Accordingly, we reasoned that LTβR stimulated non-canonical NF-κB signaling in intestinal DCs. Indeed, disruption of lymphotoxin-mediated LTβR engagement in mice using LTβR-IgG fusion protein downmodulated the non-canonical NF-κB pathway in intestinal DCs (Fig. 1E, Appendix Fig. S1E). Finally, DCs present also in the human gut, particularly in IBD patients, proficiently expressed LTβR mRNA (Fig. 7C). Previous studies suggested that LTβR modulates intestinal DC functions (doi: 10.1016/j.chom.2011.06.002). Our analyses placed the non-canonical NF-κB pathway downstream of LTβR in these DCs. Admittedly, additional genetic studies will be important to fully dissect the molecular triggers of this pathway in various intestinal cell subsets. (discussed in line #164-176, #480-484)

Dear Dr. Basak,

Thank you for the submission of your revised manuscript to The EMBO Journal and your patience during peer review. We have now received the comments of all three referees who assessed the revised version of your manuscript (their reports are included below). As you will see, all referees are satisfied with the revision, acknowledge that their previously raised concerns have been adequately addressed, and now support publication of the work.

Before we can proceed with acceptance of your manuscript, there are a few changes and corrections that we need you to address in a final version of your manuscript:

- The Figures should be removed from the main manuscript file and only be uploaded to our manuscript handling system separately as individual files. The Figure legends should remain in the manuscript, after the References list, with the appropriate headings "Figure legends" and "EV Figure legends".
- Please note that you can list no more than 5 keywords after the Abstract (you currently have 6).
- Please change the heading "Declaration of Interests" to "Disclosure and competing interests statement".
- The author contributions statement should be removed from the manuscript file. Instead, we now use CRediT to specify the contributions of each author in the journal submission system. Please feel free to use the free text box to provide more detailed descriptions during submission. See also our guide to authors for more information:
<https://www.embopress.org/page/journal/14602075/authorguide#authorshipguidelines>.
- Please change "Supplementary Information" to "Appendix" on the first page of your Appendix PDF file.
- Please change "Appendix S1-S5" to "Appendix Figure S1-S5" in the Table of Contents on the first page of your Appendix PDF file.
- The Materials and Methods need to be described in the manuscript using our "Structured Methods" format, which is now required for all research articles. According to this format, the Materials and Methods section includes a "Reagents and Tools Table" -listing key reagents, experimental models, software and relevant equipment and including their sources and relevant identifiers- followed by a "Methods and Protocols" section describing the methods using a step-by-step protocol format. The aim is to facilitate adoption of the methodologies across labs. More information on this format as well as a template (.docx) for the "Reagents and Tools Table" can be found in our author guide:
<https://www.embopress.org/page/journal/14602075/authorguide#structuredmethods>.
- Please also make sure to update accordingly your Author Checklist.
- The section "Supplemental Information" should be removed from the manuscript file.
- Please correct the section order in your manuscript as follows: title page with complete author information, abstract, keywords, introduction, results, discussion, materials and methods, data availability, acknowledgements, disclosure and competing interests statement, references, main figure legends, tables, expanded figure legends.
- Please make sure that the permanent, specific URL for each deposited dataset mentioned in the Data availability statement should be provided. Reviewer links should now be replaced by the permanent URLs, and all data should be publicly available at the time of publication.
- Please note our Data citation format (<https://www.embopress.org/page/journal/14602075/authorguide#referencesformat>) and update accordingly the data callouts in the text for datasets with identifiers "GSE148794", "GSE236531", "SCP 259", "GSE162335", "Single Cell Portal 259", and "GSE162335". All data citation callouts in the text should have matching entries in the References list (please see our guide -link above- for more information).
- Please note that the exact p values are not provided in the legends of Figures 1e, g-h, j-l; 2f, h-i; 3c-h; 4b-c, f-g; 5b, g-h; 6a-f.
- Please indicate the statistical test used for data analysis in the legends of Figures 2b-c.
- Please note that in Figures 2g; 3c-h; 5b, g-h; 6b, d; there is a mismatch between the annotated p values in the figure legend and the annotated p values in the figure file that should be corrected.
- Although "n" is provided, please describe the nature of entity for "n" in the legends of Figures 2g-h; 3a; 5b, g-h.

- Please explain in your Source Data checklist if the requested data for each Figure panel have been uploaded to our manuscript handling system or to BioStudies (accession ID: S-BSST1441). In particular, please clarify where the Source Data for Figure panels 3A-B, 3E-H, 4A, 6A-C can be accessed. You are also kindly requested to correct the labeling of Figure panels in the uploaded zip files.

- During our standard Figure checks, our Data Integrity Analyst has identified a number of issues that have to be clarified/declared in the respective Figure legends, or corrected before we can proceed with acceptance of your manuscript for publication:

1. Blot re-use between Figure 2A & Appendix Figure S1A that is not declared in the Figure legends. The re-use must be clearly stated in the respective Figure legends.
2. FACS plot re-use between Figure 4A and Appendix Figure S3A that is not declared in the Figure legends. The re-use must be clearly stated in the respective Figure legends.
3. The plots shown in the Appendix Figure S1B "DSS, d0" and "DSS, d3" appear to be identical. Please check if this is mistake and correct it, or else explain in the Figure legend.
4. Please check if the plots shown in the Appendix Figure S5A are correct.

Please also note that as part of the EMBO publications' Transparent Editorial Process, The EMBO Journal publishes online a Peer Review File along with each accepted manuscript. This File will be published in conjunction with your paper and will include the referee reports, your point-by-point response and all pertinent correspondence relating to the manuscript. You can opt out of this by letting the editorial office know (contact@embojournal.org). If you do opt out, the Peer Review File link will point to the following statement: "No Peer Review File is available with this article, as the authors have chosen not to make the review process public in this case."

We look forward to seeing a final version of your manuscript as soon as possible. Please use this link to submit your revision:

Link Not Available

Best regards,

Ioannis

Referee #1:

The authors have done an excellent job addressing the concerns raised by my review, and I feel the manuscript is acceptable in its current form.

Referee #2:

The authors have adequately addressed the points of the original review.

Referee #3:

The authors have addressed all the points I raised in the prior round. I appreciate how the study has further advanced upon revision. I have no further comments.

Please note that in the PDF, there are whole pages in which only one sentence is present. This is minor and I assume this will be resolved upon formatting of the manuscript.

Rev_Com_number: RC-2024-02368

New_manu_number: EMBOJ-2024-117451R

Corr_author: Basak

Title: Non-canonical NF- κ B signaling limits the tolerogenic b-catenin-Raldh2 axis in gut dendritic cells

The authors addressed the minor editorial issues.

Dear Dr. Basak,

Thank you for the submission of your revised manuscript to The EMBO Journal. While the majority of our previous requests have been addressed adequately, there are two remaining minor issues that we kindly ask you to address in a final version of your manuscript:

- The organization of your reagents and tools is not correct: please note that all reagents, experimental models, software and relevant equipment (including their sources and relevant identifiers) should be listed in a single table (multiple reagent tables are not allowed), which you can upload to our manuscript tracking system as a "Reagent Table" file. Please find more detailed instructions, examples, and a Word template in our author guide:
<https://www.embopress.org/page/journal/14602075/authorguide#structuredmethods>.

- Please also update all relevant callouts to the "Reagents and Tools" table throughout the manuscript.

- Please also make sure to update accordingly your Author Checklist.

- Please label the files/folders containing the Source Data that correspond to individual Figure panels with the names of those panels (for example, the Source Data for Figure panel 1A should be included in a file or a folder named "Figure 1A.xxx" within the Figure 1 Source Data zip folder).

We look forward to seeing a final version of your manuscript as soon as possible. Please use this link to submit your revision:
Link Not Available

Best regards,

Ioannis

Rev_Com_number: RC-2024-02368
New_manu_number: EMBOJ-2024-117451R1
Corr_author: Basak
Title: Non-canonical NF- κ B signaling limits the tolerogenic b-catenin-Raldh2 axis in gut dendritic cells

All editorial and formatting issues were resolved by the authors.

Dear Dr. Basak,

Congratulations on an excellent manuscript, I am very pleased to inform you that it has been accepted for publication in The EMBO Journal. Thank you for your comprehensive responses to the referee concerns and suggestions.

If you have any questions, please do not hesitate to contact the Editorial Office. Thank you for your contribution to The EMBO Journal. Working with you has been a pleasure.

Best regards,

Ioannis

Rev_Com_number: RC-2024-02368
New_manu_number: EMBOJ-2024-117451R2
Corr_author: Basak
Title: Non-canonical NF- κ B signaling limits the tolerogenic b-catenin-Raldh2 axis in gut dendritic cells